# Viral pathogens hitchhike with insect sperm for paternal transmission

Qianzhuo Mao[1,2], Wei Wu[1], Zhenfeng Liao[1], Jiajia Li[1], Dongsheng Jia[1,2], Xiaofeng Zhang[1], Qian Chen[1], Hongyan Chen[1], Jing Wei[1] & Taiyun Wei[1,2]

Arthropod-borne viruses (arboviruses) can be maternally transmitted by female insects to their offspring, however, it is unknown whether male sperm can directly interact with the arbovirus and mediate its paternal transmission. Here we report that an important rice arbovirus is paternally transmitted by the male leafhoppers by hitchhiking with the sperm. The virus-sperm binding is mediated by the interaction of viral capsid protein and heparan sulfate proteoglycan on the sperm head surfaces. Mating experiments reveal that paternal virus transmission is more efficient than maternal transmission. Such paternal virus transmission scarcely affects the fitness of adult males or their offspring, and plays a pivotal role in maintenance of viral population during seasons unfavorable for rice hosts in the field. Our findings reveal that a preferred mode of vertical arbovirus transmission has been evolved by hitchhiking with insect sperm without disturbing sperm functioning, facilitating the long-term viral epidemic and persistence in nature.

[1] Vector-borne Virus Research Center, Fujian Province Key Laboratory of Plant Virology, Fujian Agriculture and Forestry University, Fuzhou, Fujian 350002, China. [2] State Key Laboratory for Ecological Pest Control of Fujian and Taiwan Crops and College of Life Science, Fujian Agriculture and Forestry University, Fuzhou 350002, China. These authors contributed equally: Qianzhuo Mao, Wei Wu. Correspondence and requests for materials should be addressed to J.W. (email: weijing0306@163.com) or to T.W. (email: weitaiyun@fafu.edu.cn)

Many devastating plant, animal, and human pathogens are vectored by arthropod insects[1,2]. For example, *Rice stripe virus* (RSV) transmitted by planthoppers has caused a serious agricultural threat in Asian rice-growing countries[3], and Zika virus transmitted by mosquitoes has caused a recent public health threat in Americas[4]. Frequently, arthropod-borne viruses (arboviruses) can be vertically transmitted to the vector progeny population to ensure survival during adverse conditions for horizontal transmission[5–7]. Thus, vertical transmission is an important endemic maintenance mechanism for arboviruses in nature. Vertical virus transmission by insect vectors in nature may include maternal or paternal transmission[7–10]. Maternal transmission of arboviruses through transovarial passage has been extensively investigated[6,9,11–14], however, whether sperm-mediated paternal virus transmission occurs remain undetermined. Oocytes accumulate a large quantity of cytoplasm that provide a room for viral infection, whereas sperms discard their cytoplasm during spermatogenesis and transform into a streamlined shape with the small head consisting of condensed nucleus and the slender tail made of microtubule bundles for motility[15]. Therefore, if arboviruses can be paternally transmitted via insect sperm, a possible target may be the outer membrane of the sperm. Considering the extremely streamlined sperm structure, viral infection to the sperm head is expected to impair normal functioning of the sperm[16–18]. For example, the presence of human immunodeficiency virus in the human sperm and Zika virus in mice sperm would damage sperm normal functioning[16–18], and, thus, sperm-mediated paternal virus transmission may seem unlikely to occur. In this study, however, we demonstrate that a preferred mode of parental virus transmission has been evolved by hitchhiking with the sperm of male insect vectors without disturbing sperm functioning in a leafhopper-borne plant reovirus.

More than 75% of plant viruses can be transmitted by aphids, leafhoppers, planthoppers, whiteflies, and other vectors in a persistent, semi-, or non-persistent manner, thereby providing fertile ground for mechanistic studies on vector transmission[5,7,19]. The mechanisms for vertical transmission of persistent plant viruses between an infected female and its offspring through transovarial passage have been demonstrated[5,8,11–14]. For example, we have determined that RSV, a tenuivirus, and *Tomato yellow leaf curl virus*, a begomovirus, exploit the existing oocyte entry paths of vitellogenin to overcome transovarial transmission barriers in planthopper or whitefly vectors[11,12]. We also have shown recently that transovarial transmission of *Rice dwarf virus* (RDV), a plant reovirus, is mediated by the specific interaction of the viral capsid protein with the outer membrane protein of an obligate symbiotic bacterium of the vector green rice leafhoppers[20]. *Rice gall dwarf virus* (RGDV), also a plant reovirus, causes epidemic outbreaks and extensive rice yield losses in Asian rice-growing countries, and has long been thought to be transmitted by a transovarial mechanism in green rice leafhoppers[21–24]. However, we observed recently that the percentage of transovarial transmission (~ 20%) in the main vector of RGDV, the green rice leafhopper *Recilia dorsalis*, is much lower than the overall percentage (~ 80%) of vertical transmission[24]. Unlike RDV, RGDV virions encounter strong barriers to enter the oocytes in female vectors for maternal transmission[24]. The significant disparity between the percentages for the vertical and maternal transmission suggested that RGDV may have evolved to be paternally transmitted by male insects to the offspring. Here, we report that a high efficiency of sperm-mediated paternal transmission route of RGDV by male *R. dorsalis* occurs without affecting the fitness of male insects or their offspring, which may play a vital role in long-term maintenance and spread of RGDV in the field.

## Results

**Paternal transmission of RGDV through vector generations.** To explore whether RGDV can be paternally transmitted, we observed vertical transmission of RGDV from viruliferous ($V^+$) male (♂) or female (♀) *R. dorsalis* reared under controlled greenhouse conditions (Fig. 1a). In the eggs laid by the individual $V^+$♀ leafhoppers that mated with nonviruliferous ($V^-$) ♂ leafhoppers, 22% were positive for RGDV (Fig. 1b), consistent with our earlier observation[24]; in contrast, 73% were RGDV-positive of the eggs produced by $V^-$♀ leafhoppers that mated with $V^+$♂ (Fig. 1b), indicating that paternal transmission is ~ 3.3 times as efficient as maternal transmission, similar to the efficiency of vertical transmission by field-caught leafhoppers (Fig. 1c). Interestingly, the highest efficiencies of vertical transmission (82%) were observed in both laboratory-reared and field-caught leafhoppers when both parents were viruliferous (Fig. 1b, c).

We then determined the epidemiological significance of paternal virus transmission in the field. Over the past 30 years, viral disease caused by RGDV is always epidemic in the field in Southern China[7,23,24]. During the winter months (November to March) in Guangdong, Southern China when rice plants are rarely present, the weed *Alopecurus aequalis* becomes the primary habitat of rice leafhoppers for up to two generations (Fig. 1d). Generally, RGDV infection in the weed *A. aequalis* was never observed in the field during the winter months, though a very low rate of viral infection in *A. aequalis* occurred under laboratory conditions (Supplementary Table 1). Thus, the weed *A. aequalis* was not a suitable reservoir for RGDV in the field. After the late-planted rice is harvested, infected leafhoppers move to grass weeds and overwinter (Fig. 1d). The overwintering generations move to rice and spread the virus in warm areas where rice is planted in early April (Fig. 1d). We surveyed the leafhopper populations in Guangdong for the presence of RGDV in winter for 4 consecutive years. Although the percentage of viruliferous leafhoppers dropped following the transfer to *A. aequalis*, 20–30% of the overwintering leafhoppers carried RGDV in March of all 4 years when spring rice became available (Fig. 1e). Interestingly, significantly higher percentages of male leafhoppers were viruliferous than females (Fig. 1f). Paternal transmission also remained more efficient than maternal transmission when examined for three successive generations (Fig. 1g). These data indicate a pivotal role of paternal transmission in the overwintering of RGDV in the field.

**Paternal transmission does not affect offspring fitness.** We then determined whether the efficient paternal virus transmission affected the fitness of male *R. dorsalis* or their offspring. We found that $V^+$ males exhibited no significant differences in either mating (Fig. 2a) or survival (Fig. 2b) compared with $V^-$ males. Generally, one male can mate with around four virgin $V^-$ females in 3 days (Fig. 2a). Interestingly, paternal transmission of RGDV had no significant deleterious effects on female fecundity, progeny egg development and hatching rates when compared the crossing between $V^-$ female and $V^+$ male with that of $V^-$ female and $V^-$ male (Fig. 2d). In contrast, $V^+$ females died earlier than $V^+$ males and produced eggs with severe developmental defects compared with that of $V^-$ females (Fig. 2b, d). The surviving offspring from maternal transmission were able to reach adulthood and transmit the virus as efficiently as the ones from paternal transmission (Supplementary Figure 1a). Moreover, neither maternal nor paternal transmission affected the offspring sex ratios (Fig. 2c). Therefore, paternal transmission of RGDV would avoid the deleterious effect of maternal transmission on viruliferous vector population, thus promoting viral transmission. Taken together, the above data suggest that paternal transmission

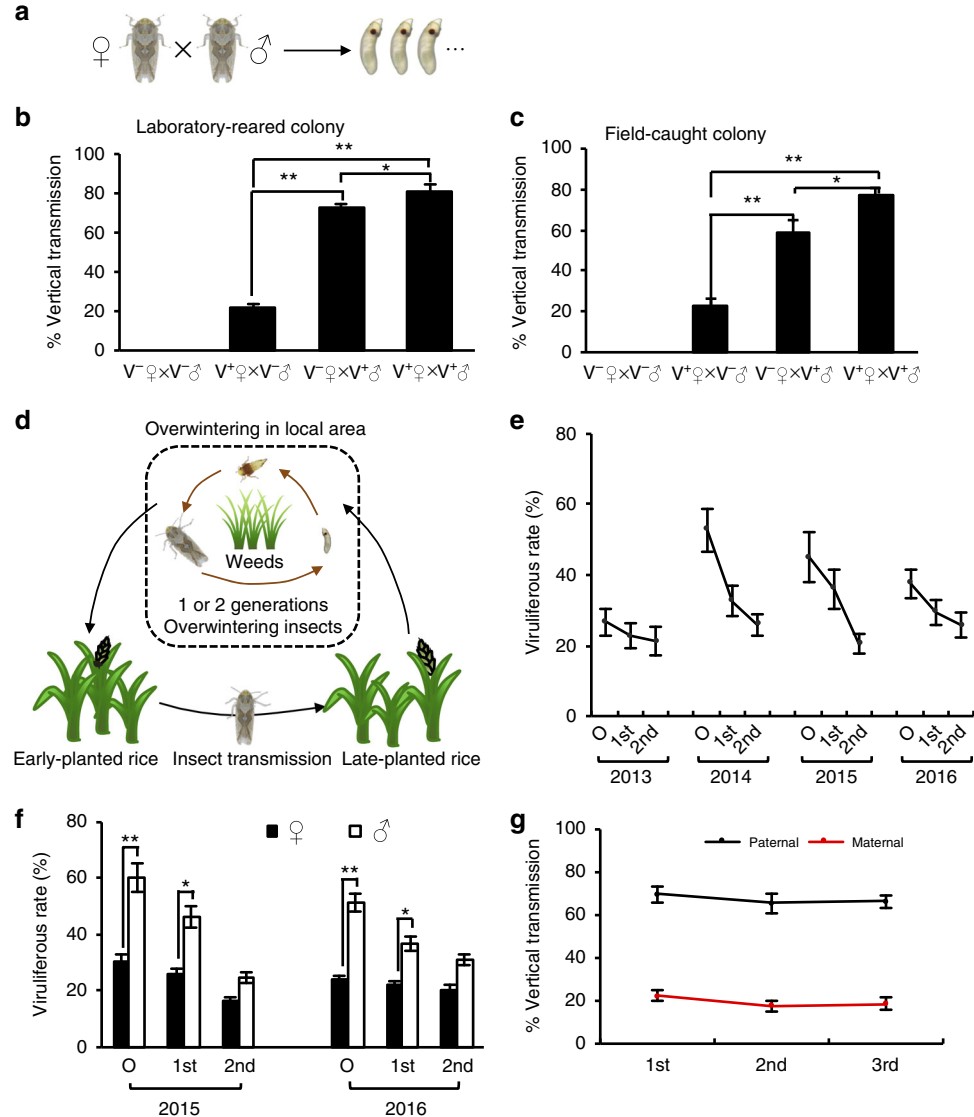

**Fig. 1** Vertical transmission of RGDV by viruliferous *R. dorsalis*. **a** Experimental design for vertical virus transmission assays. **b**, **c** Vertical transmission of RGDV by the laboratory reared **b** and the field caught **c** V+ and V− leafhoppers via mating. Eggs were collected for tracing RGDV. **d** Maintenance and epidemic cycle of rice viral disease caused by RGDV in the field in Guangdong. **e** Viruliferous rates of *R. dorsalis* populations in different generations in Guangdong during overwintering seasons from 2013 to 2016. **f** Viruliferous rates in overwintering population of male or female *R. dorsalis* in Guangdong in 2015 and 2016. **g** Successive paternal and maternal transmission rates of RGDV by *R. dorsalis* for three generations. Data are presented as mean ± SE of three independent experiments of four mating combinations in **b**, **c**, and **f**, *P < 0.05, **P < 0.01; ANOVA followed by Tukey's HSD test. Values are means ± SD of three independent experiments of four mating combinations in **e** and **g** (Student's *t* test, two tailed). V+, viruliferous. V−, nonviruliferous. **e**, **f** O, the original overwintering insects collected in November, 1st, the first generation of overwintering insects collected around January, 2nd, the second generation of overwintering insects collected in late February. **g** 1st, the first generation, 2nd, the second generation, 3rd, the third generation

of RGDV as a preferred mode of vertical transmission has been evolved during the long-term virus–vector interactions.

**Paternal transmission of RGDV by hitchhiking with the sperm.** We then investigated whether the sperm-mediated paternal transmission of RGDV occurred in male *R. dorsalis*. Our immunofluorescence and electron microscopy revealed an association of dense RGDV particles with the plasma membrane of sperm heads in the male reproductive system (Fig. 3a–f). We then used immunofluorescence microscopy to observe how RGDV hitchhiked a ride on the sperm to offspring. RGDV was initially detected in the spermatheca of V− females at 3 days post mating with V+ males (Fig. 3g–i and Supplementary Table 2). Subsequently, virus-decorated sperms in the spermatheca moved to the

oviduct for fertilizing the mature eggs during ovulation (Fig. 3j–l and Supplementary Table 2). RGDV became detectable in the genital tract but not in the oocytes of females even at 10 days post mating (Fig. 3m–o and Supplementary Table 2). Electron microscopy showed that virus-decorated sperms were present in the dissected spermatheca of V+ females (Fig. 3p), confirming the transfer of virions from V+ males to V− females. Immunofluorescence assays revealed that RGDV circulated within V− females after mating with V+ males, and ~ 63% of tested female intestines were infected at 10 days post mating (Supplementary Table 3). However, the virus was rarely observed in the oocytes, indicating that strong ovarian transmission barrier occurred (Supplementary Table 3). Moreover, V+ females, which acquired virus by mating with V+ males could transmit RGDV to rice plants, and the transmission efficiency increased with the time

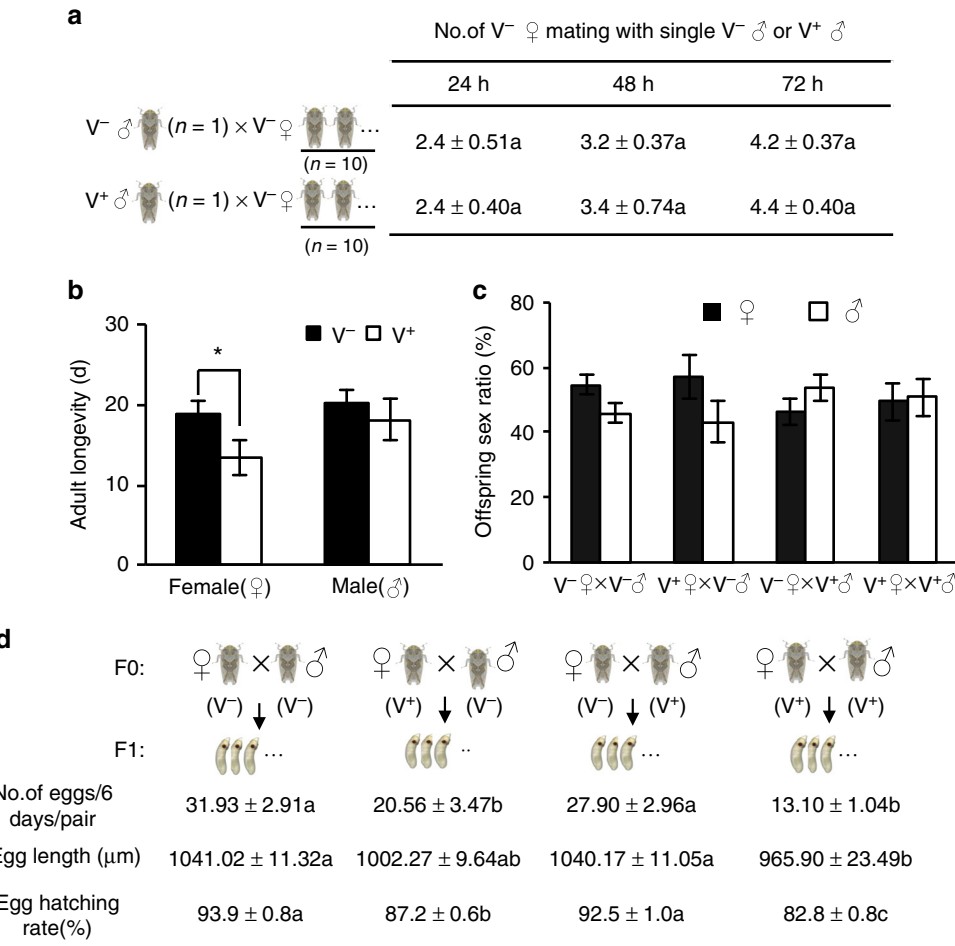

**Fig. 2** Comparison of fitness variables between male and female vectors. **a** Mean number of $V^-$ females that copulated with single $V^+$ or $V^-$ male in 24, 48, or 72 h. **b** Effects of RGDV infection on the longevity of male and female adult *R. dorsalis*. **c** Sex ratio of offspring produced by parental insects from different mating combinations ($V^-♀ × V^-♂$, $V^+♀ × V^-♂$, $V^-♀ × V^+♂$ or $V^+♀ × V^+♂$). **d** Progeny egg number, size and hatching rate of female adults from different mating combinations ($V^-♀ × V^-♂$, $V^+♀ × V^-♂$, $V^-♀ × V^+♂$ or $V^+♀ × V^+♂$). Data are presented as mean ± SE of three independent experiments of four mating combinations. The significance of any differences was tested using Student's *t* test **a–b** or Tukey's HSD test **c–d**. $^*P < 0.05$. Different letters after means in the same column **a** or line (**d**) indicate a significant difference at $P = 0.05$, and the means do not differ significantly if they are indicated with the same letter. $V^+$, viruliferous. $V^-$, nonviruliferous

post mating (Supplementary Figure 1b). These observations provide direct evidence for the paternal transmission of RGDV by "hitchhiking" with the sperms in the *R. dorsalis* male reproductive system to the spermatheca of females, from there the sperms move to the oviduct to fertilize the eggs passing through outwards to be deposited (Fig. 3q).

**Interaction of RGDV P8 and heparan sulfate proteoglycan**. We further determined how RGDV virions were associated with the plasma membrane of sperm heads. We detected the initial binding of purified RGDV virions to the head of live sperm dissected from nonviruliferous *R. dorsalis* after 5 min incubation (Fig. 4a). With longer incubation times, more virions accumulated on the sperm heads (Fig. 4a). Similar in vitro binding of RGDV to live sperms was also detectable after incubation with the major outer capsid protein P8, but not with the minor outer capsid protein P2 of RGDV (Fig. 4b, c and Supplementary Figure 2a, b). Furthermore, the specific binding to the head of live sperms was abolished by pretreatment with antibodies against intact virions or P8, but not P2 (Fig. 4d). These results indicate that the major outer capsid protein P8 mediates the direct binding of RGDV virions to the sperm head.

To identify the sperm attachment protein, RGDV P8 fused to glutathione-*S*-transferase (GST) was generated as bait to screen the sperm proteins extracted from *R. dorsalis* (Supplementary Figure 3a). Mass spectrometry analysis of the pulled down proteins identified 12 peptides with four different types mapped to heparan sulfate proteoglycan (HSPG) (Supplementary Figure 3b), a ubiquitous cell surface protein, which has been exploited by many pathogens such as viruses, bacteria, and parasites for their initial attachment and subsequent cellular entry[25,26]. Here, we determined that *R. dorsalis* HSPG was 3941 amino acids long and consisted of five domains (GenBank accession number MH060173) (Fig. 4e). The 12 identified peptides all targeted the third domain (domain III) of HSPG (Supplementary Figure 3b). Yeast two-hybrid assay showed that RGDV P8 interacted only with the domain III of HSPG (Fig. 4f and Supplementary Figure 4a–f). The specific interaction between RGDV P8 and HSPG domain III was also independently verified by GST pull-down assay (Fig. 4g). RT-qPCR and western blot assays revealed an enriched expression of HSPG in the male reproductive system compared with the remaining tissues or the female reproductive system of *R. dorsalis* (Supplementary Figure 5a–e). Specific HSPG antibodies recognition of the sperm head and testis, but not the intestines, was also verified by immunofluorescence microscopy (Supplementary

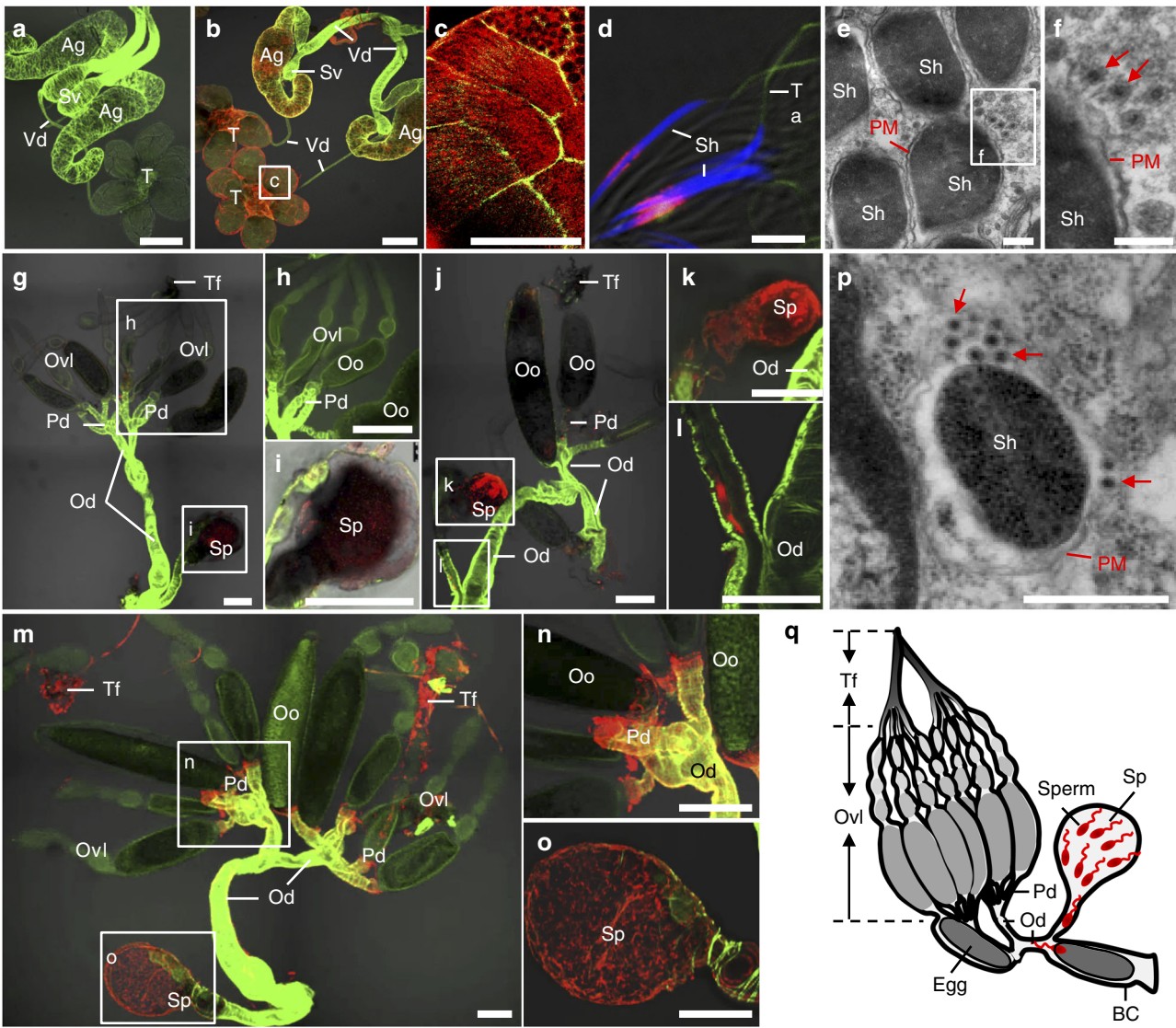

**Fig. 3** Sperm-mediated paternal transmission of RGDV from V+ males to offspring. **a–c** Immunofluorescence microscopy showing RGDV infection in V−
**a** or V+ male reproductive systems **b**, **c**. **c** is the enlargement of boxed area in **b**. The male reproductive systems were stained with virus-rhodamine (red)
and actin dye Phalloidin-FITC (green). Bars: **a**, **b**, 200 μm; **c**, 50 μm. **d** Immunofluorescence microscopy showing association of RGDV virions with sperm
heads. Sperms from V+ male were stained with virus-rhodamine (red), actin dye Phalloidin-FITC (green) and DAPI (blue). Bar, 10 μm. **e**, **f** Electron
micrograph showing association of RGDV virions (red arrows) with the plasma membrane of sperm heads. **f** is the enlargement of boxed area in **e**. Bars,
200 nm. **g–o** Immunofluorescence microscopy showing infection route of RGDV in V− female reproductive system after mating with V+ males. RGDV
accumulated in the spermatheca **g** and **i** but not in other parts **g** and **h** at 3 days post mating. **h** and **i** are the enlargements of boxed areas in **g**. **j–l** RGDV
accumulation in the opening of the spermatheca **k** where it connected to the oviduct **l** at 6 days post mating. **k** and **l** are the enlargements of boxed areas in
**j**. **m–o** RGDV accumulation in the spermatheca **o** but not in female ovaries **n** at 10 days post mating. **n** and **o** are the enlargements of boxed areas in **m**.
Bars, 200 μm. **g–o** The female reproductive systems were stained with virus-rhodamine (red) and actin dye Phalloidin-FITC (green). **p** Electron microscopy
showing association of RGDV virions (red arrows) with the plasma membrane of sperm heads in the spermatheca of V+ females at 10 days post mating.
Bar, 100 nm. **q** Proposed model for sperm-mediated paternal transmission of RGDV. During mating, virus-associated sperms accumulate first in the female
spermatheca, then move to the oviduct to fertilize eggs. All immunofluorescence images are representative of at least three replicates. AG, accessary
gland; BC, bursa copulatrix; Od, oviduct; Oo, oocyte; Ovl, ovariole; Pd, pedicel; PM, plasma membrane; Sh, sperm head; Sp, spermatheca; SpG, spermatheca
gland; Sv, seminal vesicle; T, testis; Tf, terminal filament; Vd, vas deferens

Figure 5f). We found that the expression of HSPG in the male
reproductive system was significantly induced in viruliferous
leafhoppers (Fig. 4h, i). Furthermore, the colocalization of RGDV
and HSPG on the head of sperm was readily detectable (Fig. 4j)
and sperm binding by RGDV virions was strongly reduced by
pretreatment with purified HSPG-specific antibodies (Fig. 4k).
Importantly, microinjecting dsRNA targeting HSPG mRNA into
the newly emerged viruliferous male *R. dorsalis* knocked down the
in vivo expression of HSPG, thereby inhibited the sperm binding

by RGDV virions (Fig. 4l, m and Supplementary Figure 6), and
reduced the accumulation of RGDV in the male reproductive
systems (Fig. 4n) and the subsequent paternal transmission of
RGDV to offspring (Fig. 4o). It is clear that RGDV infection can
trigger the enriched accumulation of HSPG in vector male
reproductive system to benefit its paternal transmission. Taken
together, these data indicate that the specific binding of RGDV
virions to the sperm head mediated by P8-HSPG interaction has a
key function in the paternal vertical transmission (Fig. 4p).

**Paternal transmission of RGDV by minor leafhopper vector**. In addition, we examined if this phenomenon also occurred in minor vector of RGDV, the rice leafhopper *Nephotettix cincticeps*. We first observed that the viruliferous rates of natural *R. dorsalis* and *N. cincticeps* population in Guangdong were ~ 58% and 5% during planting seasons, respectively (Fig. 5a). The acquisition efficiency of *N. cincticeps* under experimental conditions was only 13%, whereas that of *R. dorsalis* was over 80% (Fig. 5b). In the eggs laid by the individual V⁺ female that mated with V⁻ male of

*N. cincticeps*, 18% were positive for RGDV (Fig. 5c); in contrast, 62% were RGDV-positive of the eggs produced by V⁻ female that mated with V⁺ male of *N. cincticeps*, indicating that paternal transmission is ~ 3.4 times as efficient as maternal transmission by *N. cincticeps*, similar to the efficiency of paternal transmission by *R. dorsalis* (Fig. 5c). Immunofluorescence microscopy also showed a close association of RGDV with the head surface of sperms dissected from seminal vesicles of V⁺ males of *N. cincticeps* (Fig. 5d). However, virus attachment to the sperm of the

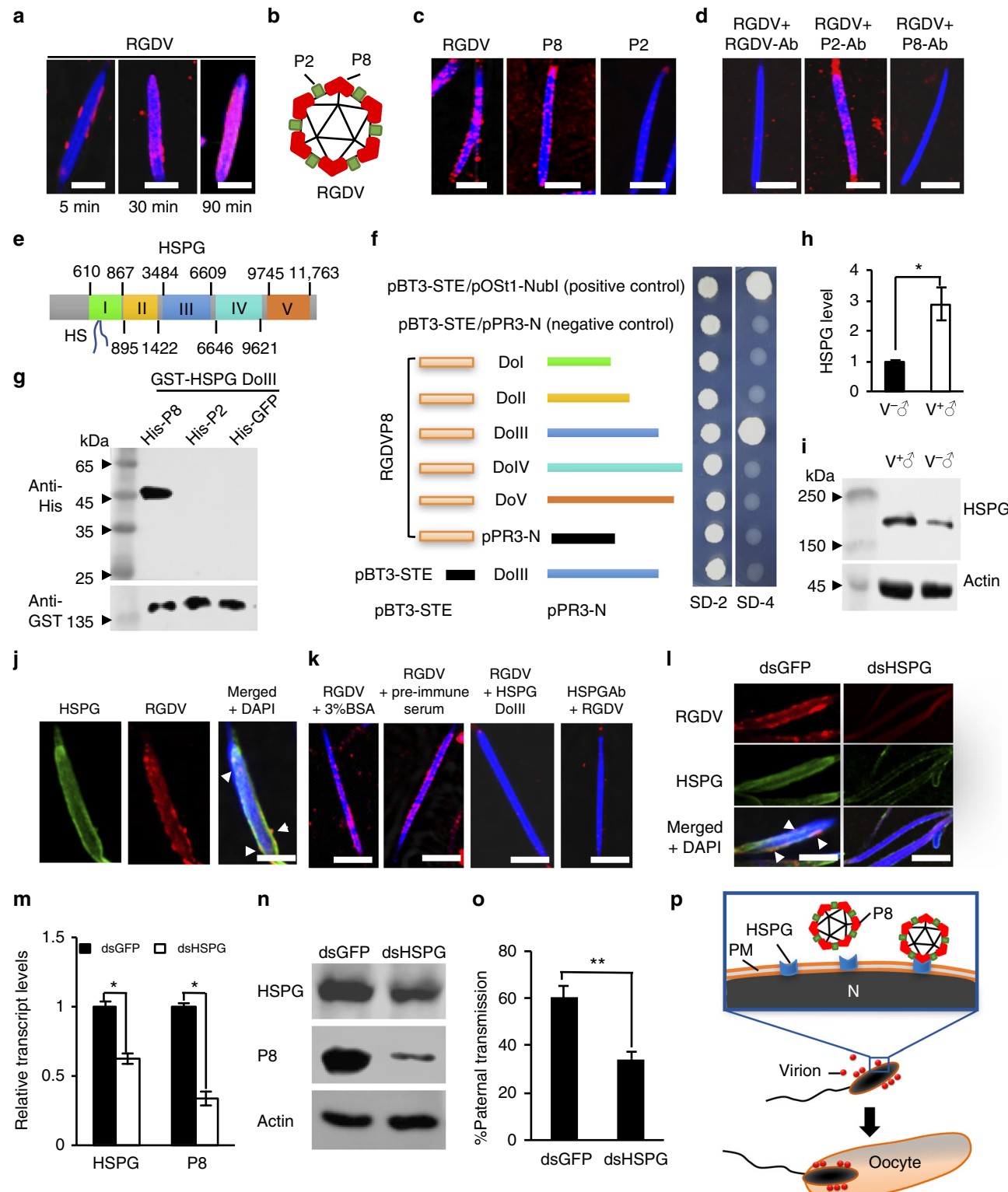

non-vector of RGDV, the wheat leafhopper *Psammotettix alienus*, was not observed (Fig. 5d). Moreover, the amino-acid sequences of HSPG domain III between the two rice leafhoppers were only two amino acids difference, but they differed in ~ 40 amino acids from that of the non-vector *P. alienus* (Fig. 5e). A yeast two-hybrid assay also indicated that RGDV P8 specifically interacted with HSPG domain III of *N. cincticeps*, rather than with that of *P. alienus* (Fig. 5f). Thus, exploiting HSPG for the efficient parental transmission of RGDV is a conserved mechanism in rice leafhopper vectors.

## Discussion

A handful of studies have described the mechanistic basis of vertical maternal transmission of arboviruses and its role in epidemic persistence of viruses[6–8], whereas the occurrence of vertical paternal transmission in nature is undetermined. In this study, we discovered a previously unknown phenomenon: a rice arbovirus can be efficiently transmitted from male insect vectors to offspring through a direct interaction of viral outer protein with the cell surface HSPG of the sperm head without affecting the fitness of male insects or their offspring. More importantly, an infected male can potentially transfer viruses to more offspring because a male can mate repeatedly with females and thus enhance virus spread. By contrast, during maternal transmission, viral propagation in the oocytes of female ovary often causes cytopathologic changes, decreasing the fitness of insect offspring[24,27]. Thus, males transmit a remarkably higher proportion of viruses to insect offspring through sperms than females transmit through ovaries. Furthermore, viruliferous males survive much longer than viruliferous females, and finally, more males are infected by RGDV than females in the field. More importantly, the paternal transmission rate (~ 60%) is evidently higher than the maternal rate (~ 20%) in field-collected *R. dorsalis* populations. Thus, during the cold seasons unfavorable for virus-infected rice hosts in the field, the maintenance efficiency of RGDV through up to two insect generations tends to decrease. We deduce that such a sperm-mediated paternal transmission is a more powerful type of vertical virus transmission than maternal transmission by insect vectors in many cases, and plays a vital role in the efficient maintenance of RGDV during the cold seasons in the field. Taken together, a preferred mode of vertical arbovirus transmission has been evolved by hitchhiking with insect sperm, thereby explaining a natural long-term endemic pattern of RGDV throughout Southern China for > 30 years.

At present, whether other vector-borne viral pathogens can be carried by insect sperms and then be paternally transmitted with high efficiency in nature is as yet unknown. Interestingly, arboviruses such as La Crosse virus and Zika virus can be paternally transmitted by male mosquitoes with low efficiency, but viral antigens were not observed within sperms[9,28–30]. It seems that La Crosse virus and Zika virus in mosquitoes can be venereally transmitted by male accessory sex gland fluid rather than by sperm, which agrees with the findings for the symbiotic rhabdoviruses in *Drosophila*[31]. We cannot rule out the possibility that paternal transmission of RGDV occurs not through sperm but through seminal fluid. However, even though RGDV can enter into female oocytes from the infected male seminal fluid, it would encounter strong ovarian barriers[24], and, thus it appears to be of a low level of occurrence and constitute a relatively minor component of the vertical transmission rate. Given the widespread occurrence of long-term epidemic for RGDV in nature, the virus has gained an evolutionary advantage by hitchhiking with the insect sperm for vertical propagation. We anticipate other arboviruses may have also evolved a similar strategy to ensure viral survival during adverse conditions for horizontal transmission.

Our study reveals a mode of vertical transmission of an arbovirus. Lack of virus-infected host plants in cold seasons presents a bottleneck for arbovirus transmission so that sperm-mediated paternal transmission in vector populations may be critical to viral persistence in the field. Our data lay a foundation for further investigation of the mechanisms underlying paternal virus transmission and ecological significance, and provide insights into the development of efficient approaches to attenuate viral epidemic by targeting sperm-mediated paternal transmission mechanism.

## Methods

**Crossing experiments**. Four treatments of crossing were conducted for *R. dorsalis* adults from a laboratory-reared colony and a field-caught colony, respectively. In each treatment, one newly emerged adult female was crossed with one newly emerged adult male as follows: (i) V⁻ female (V⁻♀) × V⁻ male (V⁻♂); (ii) V⁺ female (V⁺♀) × V⁻ male (V⁻♂); (iii) V⁻ female (V⁻♀) × V⁺ male (V⁺♂); and (iv) V⁺ female (V⁺♀) × V⁺ male (V⁺♂). The V⁻ and V⁺ *R. dorsalis* populations have been established in our laboratory (Supplementary Materials and Methods). Our preliminary test confirmed that > 80% of insects contained RGDV in the viruliferous populations[27]. Thus, for laboratory-reared colony, the 5th instar nymphs were caught from V⁻ or V⁺ *R. dorsalis* populations and reared separately in glass tubes until eclosion for crossing. Generally, the percentage of V⁺ *R. dorsalis* populations caught from the areas where RGDV incidence was high was above 60%. Thus, for field-caught colony, the individual 3rd or 4th instar nymphs of *R. dorsalis* were caught directly from the rice fields where RGDV incidence was high or from virus-free rice fields in Guangdong, China and reared separately in glass tubes until eclosion. For the mating procedure, the newly emerged potential V⁺ adults were chosen for mating one to one with the V⁻ adults in glass tubes containing one rice seedling for 5 days, and the seedling was replaced daily with a new one to avoid viral acquisition from plant hosts. At the end of the fifth day, the potential V⁺ males in each of the tubes were collected for virus detection by real-time-polymerase chain reaction (RT-PCR) assay, and the females were left in each of the tubes to lay eggs for another 5 days and the seedling in each tube was renewed daily. The potential V⁺ females were then collected for virus detection by RT-PCR assay. At 7 days after the removal of females, the seedlings were dissected to collect eggs for RGDV detection by RT-PCR assay. For each of the four mating types in the three independent experiments with the laboratory-reared colony, the number of eggs detected in each independent experiment was as follows: V⁻♀ × V

**Fig. 4** Interaction of RGDV P8 with HSPG. **a** Live sperms were incubated with purified virions for 5, 30, or 90 min, then stained with virus-rhodamine (red) and DAPI (blue). **b** Structure of RGDV virion. **c** Live sperms were incubated with purified virions, P8 or P2, then stained with virus-, P8-, or P2-rhodamine (red) and DAPI (blue). **d** Live sperms were incubated with purified virions and RGDV-, P8-, or P2-specific antibodies, then stained with virus-rhodamine (red) and DAPI (blue). **e** Schematic representation of *R. dorsalis* HSPG gene. **f** Interactions between different domains of HSPG and RGDV P8 in the yeast two-hybrid system. SD-2: SD-Trp-Leu, SD-4: SD-Trp-Leu-His-Ade. **g** Pull-down analysis of RGDV P8 interaction with HSPG DoIII. HSPG DoIII was fused with GST as a bait protein. P8 was fused with His as a prey protein. P2 and GFP were fused with His as controls. **h, i** The transcript **h** or protein **i** levels of HSPG in reproductive systems of V⁻ or V⁺ male adults. **j** Confocal micrographs showing colocalization (white triangles) of RGDV (red) and HSPG (green) on sperm head (blue) surface. **k** Live sperms were incubated with purified virions and 3% BSA, pre-immune serum, purified HSPG DoIII, or HSPG-specific antibodies, then stained with virus-rhodamine (red) and DAPI (blue). **l** The reduced colocalization (white triangles) of RGDV (red) and HSPG (green) on the surface of sperm heads (blue) after dsHSPG treatment. **m, n** The transcript **m** or protein **n** levels of HSPG and RGDV P8 in male reproductive systems after dsGFP or dsHSPG treatments. **o** Paternal transmission rates of RGDV from crosses between dsHSPG- or dsGFP-treated V⁺ males with V⁻ females. Eggs were collected for tracing RGDV. **p** Proposed model for interaction of RGDV P8 with HSPG mediating viral attachment on the surface of sperm heads and subsequent paternal transmission. All images are representative of at least tree replicates. **h, m**, and **o** Data are presented as mean ± SD from three independent experiments (Student's *t* test, two tailed); *P < 0.05, **P < 0.01. V⁺, viruliferous. V⁻, nonviruliferous. Ab, antibodies. PM, plasma membrane. N, nucleus. Bars, 5 μm

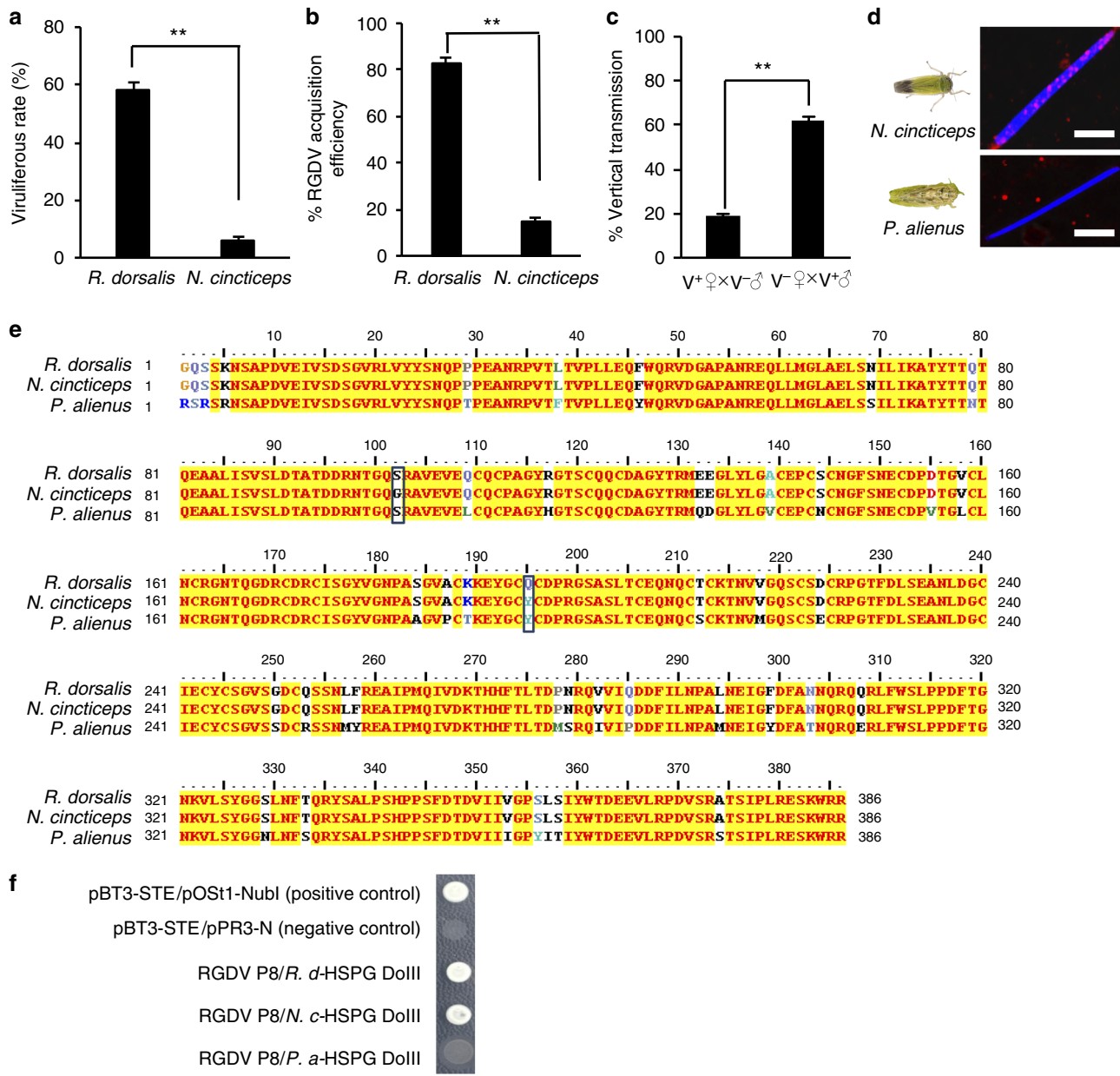

**Fig. 5** Paternal transmission of RGDV by minor vector, the rice leafhopper *N. cincticeps*. **a** The viruliferous rates of *R. dorsalis* and *N. cincticeps* (25 males and 25 females) collected from the fields in Guangdong, China. **b** The acquisition efficiencies of RGDV by *R. dorsalis* and *N. cincticeps* collected from the fields. **c** Vertical transmission of RGDV by the field-caught V⁺ and V⁻ *N. cincticeps* via mating. Eggs were collected for tracing RGDV. **d** Association of RGDV particles with sperm heads of *N. cincticeps* and the non-vector wheat leafhopper *P. alienus*. The sperms dissected from the male leafhoppers *N. cincticeps* or *P. alienus* were microinjected with RGDV virions, stained with virus-rhodamine (red) and DAPI (blue), and examined by immunofluorescence microscopy. All images are representative of at least three replicates. Bars, 10 μm. **e** Comparison of amino-acid sequences of HSPG DoIII of *R. dorsalis*, *N. cincticeps,* and *P. alienus*. **f** A yeast two-hybrid assay was used to detect the interaction between RGDV P8 and HSPG DoIII from *R. dorsalis* (*R. d*), *N. cincticeps* (*N. c*) and *P. alienus* (*P. a*). **a–c** Data are presented as mean ± SD from three independent experiments (Student's *t* test, two tailed), **$P < 0.01$

⁻♂, $n_1 = 51$, $n_2 = 65$, $n_3 = 69$; V⁺♀ × V⁻♂, $n_1 = 72$, $n_2 = 72$, $n_3 = 76$; V⁻♀ × V⁺♂, $n_1 = 78$, $n_2 = 77$, $n_3 = 81$; V⁺♀ × V⁺♂, $n_1 = 51$, $n_2 = 62$, $n_3 = 72$. For each of the four mating types in the three independent experiments with the field-caught colony, the number of eggs detected was as follows: V⁻♀ × V⁻♂, $n_1 = 66$, $n_2 = 66$, $n_3 = 50$; V⁺♀ × V⁻♂, $n_1 = 65$, $n_2 = 66$, $n_3 = 55$; V⁻♀ × V⁺♂, $n_1 = 78$, $n_2 = 77$, $n_3 = 81$; V⁺♀ × V⁺♂, $n_1 = 55$, $n_2 = 62$, $n_3 = 84$.

**Immunofluorescence staining**. For visualizing viral infection to the male reproductive system, second instar nymphs of *R. dorsalis* were fed on diseased rice plants for 2 days and then transferred to non-infected rice seedlings. At different days after eclosion, the reproductive system from 30 males was excised, fixed, immunolabeled with virus-specific IgG conjugated to rhodamine (virus-rhodamine) (0.5 μg/μl) and actin dye phalloidin-fluorescein isothiocyanate (FITC) (Invitrogen, cat. F432; 1:200), and then processed for immunofluorescence microscopy. For

visualizing viral association with sperm, mature sperms were excised from the seminal vesicles of V⁺ males and then smeared on poly-lysine-treated glass slides. The sperms were fixed and immunolabeled with virus-rhodamine (0.5 μg/μl), and then stained with 4′,6-diamidino-2-phenylindole (DAPI) (Sigma, cat. D9542). For visualizing the virus in females after transfer from males, virgin V⁻ female adults were mated one on one with V⁺ male adults in individual glass tubes for 3 days. Males were then collected and confirmed for RGDV-positive by RT-PCR assay. The reproductive system was excised from each of the 30 female adults at 3, 6, and 10 days after mating with the V⁺ males, and then immunolabled with virus-rhodamine (0.5 μg/μl) and phalloidin-FITC (Invitrogen, cat. F432, 1:200) to visualize the route of viral transfer.

**Electron microscopy**. The seminal vesicles of V$^+$ male adult *R. dorsalis* or the spermatheca of female adult *R. dorsalis* at 3 days after crossing with V$^+$ males were excised, fixed with 2% v/v glutaradehyde and 2% v/v paraformaldehyde in posphate-buffered saline (PBS) for 2 h at room temperature, and then postfixed with 1% w/v osmium tetroxide in PBS for 1 h at room temperature. The fixed samples were dehydrated in grade series of ethanol up to 100% and embedded in Spurr's resin (SPI Ltd). Sections were observed with a transmission electron microscope (H-7650, HITACHI).

**RGDV-sperm binding in vitro**. Mature, live sperms from the seminal vesicles were smeared on poly-lysine-treated glass slides, and then incubated with purified RGDV virions (0.01 μg/μl) for 5, 30, or 90 min For detecting the binding of P2 or P8 of RGDV with sperm in vitro, His-tag-fused P2 or P8 was expressed in *Escherichia coli* strain *Rosetta*, and the proteins were purified using nickel-nitrilotriacetic acid resin (Qiagen). Sperm smears were also incubated with the purified proteins (0.5 μg/μl) for 60 min and then stained with P2- or P8-specific IgG conjugated to rhodamine, P2-rhodamine (0.5μg/μl) or P8-rhodamine (0.5 μg/μl), respectively, and then processed for confocal microscopy. Alternatively, purified RGDV virions were preincubated with P2 or P8 antibodies (0.5 μg/μl) for 10 min, and then incubated with live sperms. The samples were stained with virus-rhodamine and DAPI, and then processed for confocal microscopy.

**Identification of sperm proteins that interact with RGDV P8**. RGDV P8 was used as bait to screen the interacting proteins extracted from sperms of *R. dorsalis* using a GST pull-down assay. In brief, the GST-fused P8 or GST was bound to GST-Sepharose 4B beads (GE) for 3 h at 4 °C. Sperms were collected from the seminal vesicles of the males, suspended in PBS and lysed by ultrasounds. Sperm lysates were then centrifuged at 15,000×*g* for 15 min, and the supernatants were incubated with GST or GST-P8 conjugated Sepharose beads for 4 h at 4 °C. After extensive wash of the column, bound proteins were eluted, and then the eluted proteins were treated with dithiotreitol (1 mM) for 30 min followed by alkylation with iodoacetamide (50 mM) for 30 min The mixture was incubated with trypsin (1 μg/50 μg protein) (Promega) overnight, centrifuged at 14,000 × *g* for 2 min and the pellets were collected. Peptides were eluted into autosampler vials with 50 μl buffer (84% ACN in 0.5% acetic acid). Organic solvent was removed in a SpeedVac concentrator and the final sample volume was adjusted with buffer (2% ACN in 0.1% TFA) to 12 μl. A nanoflow high performance liquid chromatography instrument (Thermo Fisher Scientific) was coupled on-line to a Q Exactive (Thermo Fisher Scientific) with a nanoelectrospray ion source (Proxeon). The peptide mixture (5 μg) was loaded onto a C18-reversed phase column (2-cm long, 100 μm inner diameter) and separated with a linear gradient of 4–100% buffer B (80% ACN and 0.5% acetic acid). Data processing was performed using MaxQuant v1.6.0.1 software. The experiment was replicated three times. Peptides identified specifically from GST-P8 bound proteins were selected for analysis. Around 12 peptides targeted HSPG were identified in each replicate. We then used PCR, 5′ and 3′ RACE approaches to clone the full-length HSPG cDNA based on the transcriptome database of *R. dorsalis*. It was a unique gene in the transcriptome database of *R. dorsalis*. The full-length cDNA (13,267 bp) of HSPG was obtained, and it contained five domains (I–V) (Supplementary Figure 3).

**Yeast two-hybrid assay**. Interaction between HSPG of *R. dorsalis*, *N. cincticeps* or *P. alienus* and the major capsid protein P8 of RGDV was tested in a yeast two-hybrid assay (Y2H) using a DUALmembrane starter kit (DUALsystems Biotech) according to the manufacturer's manual. The P8 gene of RGDV was cloned into the bait vector pBT3-STE (PBT-STE-RGDV P8), and the five domain sequences of HSPG of *R. dorsalis* as well as HSPG domain III of *N. cincticeps* and *P. alienus* were cloned into the prey vector pPR3-N (pPR3-N-DoI, pPR3-N-DoII, pPR3-N-DoIII, pPR3-N-DoIV, and pPR3-N-DoV) (Supplementary Table 4). The bait and prey plasmids were co-transformed into the yeast strain NMY51. Plasmids pBT3-STE-RGDV and pPR3-N, pBT3-STE and pPR3-N-DoIII were co-transformed to test self-activation. Plasmids pBT3-STE and pOST1-Nubl (positive control) or pPR3-N (negative control) were used to cotransform NMY51. All transformants were grown on synthetic dropout (SD)-Trp-Leu agar plates (SD-2) and SD-Trp-Leu-His-Ade agar plates (SD-4) for 3–4 days at 30 °C. To assay the expression of fusion proteins, each yeast extract of tranformant strain was tested by western blot using an antibody against the LexA domain of the bait fusion protein (DUALsystems Biotech, cat. P06004, 1:500) or an antibody against HA Tag domain (DUALsystems Biotech, cat. P06005, 1:500) of the prey fusion protein.

**Pull-down assay**. A GST pull-down assay was used to confirm the interaction between the protein encoded by the domain III fragment of the HSPG gene and the P8 gene of RGDV. In brief, the domain III sequence of the HSPG gene was cloned into PGEX-3 for fusion with GST. The P8 gene of RGDV, P2 gene of RGDV and the GFP gene were cloned into pBT30a for His-tag fusion (Supplementary Table 4). All recombinant proteins were expressed in *E. coli* strain BL21. GST-HSPG-DoIII was incubated with glutathione-Sepharose beads (Amersham) for 4 h at 4 °C, then the mixture was centrifuged for 5 min at 500 g, and the supernatant was discarded.

His-P8, His-P2, or His-GFP was added to the beads, and the mixture was incubated for 4 h at 4 °C. The beads were collected and washed with washing buffer (300mM NaCl, 10 mM Na$_2$HPO$_3$, 2.7 mM KCl and 1.7 M KH$_2$PO$_4$). Immunoprecipitated proteins were detected by western blot with His-tagged antibodies (Sigma, cat. SAB4301134, 1:1000) and GST-tagged antibodies (Sigma, cat. SAB4200692, 1:1000) separately.

**Knockdown of HSPG expression in *R. dorsalis***. A DNA fragment spanning a 774-bp segment targeting HSPG domain III was amplified by PCR using forward primer 5′-*ATTCTCTAGAAGCTTAATACGACTCACTATAGG*GGGCGTGCGAC TGGTGTACTA-3′ and reverse primer 5′-*ATTCTCTAGAAGCTTAATACGACT CACTATAGGG*GAGTGTGAAGTGGTGCGTCTTG-3′, both possessing a T7 promoter (shown in italic) at the 5′ end. PCR products were used for the synthesis of dsRNAs targeting HSPG domain III (dsHSPG) according to the protocol for the T7 RiboMAX Express RNAi System kit (Promega). The 630-bp segment of the GFP gene was used for in vitro dsRNA syntheses (dsGFP) as the control (Supplementary Table 4)[32]. Second instar nymphs of *R. dorsalis* were allowed to feed on RGDV-infected rice plants for 2 days, and then transferred to non-infected rice plants. Newly emerged male adults were microinjected with 0.2 nl dsRNAs (0.02 μg/μl) with a Nanoject II Auto-Nanoliter Injector (Spring), and then placed on non-infected rice seedlings. Different concentrations of the dsHSPGs were microinjected into newly emerged male adult leafhoppers, and the mortality of insects was assessed every day. For each concentration, 30 insects were microinjected, and three replicates were performed. To measure the effects of dsRNA on the transcript levels of the P8 gene of RGDV and HSPG gene of *R. dorsalis*, the male reproductive systems were excised from dsRNA-treated male adults at 5 days after microinjection, and then total RNAs were extracted using Trizol reagent (Invitrogen). The transcript levels of the P8 gene of RGDV and HSPG gene were quantified by relative RT-qPCR assay with the SYBR Green PCR MasterMix kit (Promega) in a Mastercycler Realplex4 real-time PCR system (Eppendorf) in accordance with the manufacturer's instructions. Relative levels of gene expression were normalized to a housekeeping gene *elongation factor 1alpha* gene (EF1, accession number AB836665) and estimated by the $2^{-\triangle\triangle Ct}$ (cycle threshold) method. A pool of 20 insects was used for each replicate, and the experiment was replicated three times. Furthermore, the total proteins were extracted from dissected male reproductive systems of 40 dsRNA-treated leafhoppers. P8 and HSPG protein levels were analyzed by western blot assay using P8- and HSPG-specific IgGs, respectively. To determine whether dsHSPG treatment inhibited paternal transmission, at 3 days after microinjection, one dsHSPG- or dsGFP-treated V$^+$ virgin male was placed with one V$^-$ virgin female in a glass tube containing a rice seedling for 3 days. Ten pairs were performed for each treatment. The male leafhoppers were then assayed for RGDV, and female leafhoppers were left in the tubes for oviposition. The offspring of each cross (dsGFP-treated, $n = 136$; dsHSPG-treated, $n = 111$) were tested for RGDV by RT-PCR assay.

**Statistical analyses**. All data were analyzed with SPSS (version 17.0; SPSS, USA). Percentage data were arcsine square root-transformed before analysis. Multiple comparisons of the means were conducted using a one-way analysis of variance followed by Tukey's honestly significant difference test at the $P<0.05$ significance level. Comparisons between two means were conducted using Student's *t* test. The data were back-transformed after analysis in the text, figures, and tables.

**Reporting summary**. Further information on experimental design is available in the Nature Research Reporting Summary linked to this article.

## Data availability

The authors declare that all data supporting the findings of this study are available in the manuscript and its Supplementary Information files are available from the corresponding authors upon request.

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

## Acknowledgements

We are grateful to Shouwei Ding for helpful discussions and editorial assistance, to Shusheng Liu for comments on the manuscript. We thank Dr. Xifeng Wang from Chinese Academy of Sciences for providing the *P. alienus* colony. This work was supported by grants from National Natural Science Foundation of China (Grants 31730071, 31770166, 31870148, and 31870149) and National Key Research and Development Plan Foundation (2016YFD0300707).

## Author contributions

Q.M., W.W. and T.W. designed all experiments. Q.M., Z.L., J.L., X.Z., Q.C. and H.C. performed crossing experiments, field investigation, and fitness measurement experiments. Q.M., Z.L. and D.J. performed the experiments for immunofluorescence staining and electron microscopy. Q.M., W.W. and Z.L. performed the protein interaction experiments and found the role of HSPG. Q.M., W.W. J.W. and T.W. analyzed the data. T.W. organized the project and wrote the manuscript. All authors read and approved the manuscript.

## Additional information

**Competing interests:** The authors declare no competing interests.

