## [Peer Review File · Nature Communications]

Reviewers' comments:

Reviewer #1 (Remarks to the Author):

This manuscript by Mao et al provided a definitive and mechanistic insight into one of the most perplexing and puzzling mechanisms of arbovirus transmission. While the subject has been known for some time now, and several proposed methods have been put forward over the years (most revealing experiments done with transovarial transmission of LaCrosse virus), this is the first manuscript that presents in an insightful and elegant way the mechanism by which paternal transmission occurs. Both methods of transmission (venereal as well as transovarial) have far reaching consequences since they have been proposed for many years as the most plausible mechanisms to explain the maintenance of arboviruses in the interepidemic years of transmission. This certainly may be a potential and highly effective method (paternal) of transmission that could also explain the transmission of insect specific arboviruses, a diverse group of viruses that have been discovered and described in some detail over the past 6-7 years.

It is quite remarkable that the model plant viruses used herein utilize heparin sulfate moieties to attach on the outer sperm surface, suggesting a conserved mechanism as HS has been identified as a receptor for rapid penetration of viruses, and bacteria on their target cells.

Overall this is a seminal and groundbreaking study that will impact the field of arbovirology for some years to come. One remaining aspect that this work barely touched is how frequent this mode of transmission is in nature which is urgently needed to inform dynamic models of transmission.

The manuscript is well written and can only require a handful of cosmetic corrections;

line 215-216. Please cite the studies describing mechanisms of arbovirus maternal transmission

Reviewer #2 (Remarks to the Author):

Mao et al. introduce a novel concept and evidence for the paternal transmission of the RGDV and their impact on the virus and host survival as well as the epidemiological relevance of such mode of transmission. The authors show compelling evidence of vector-host interactions resulting in the

paternal transmission of the RGDV as the most efficient method for its maintenance in nature. This transmission mechanism was also found in the minor RGDV vector *N. cincticeps* while no attachment of the virus to the sperm of a non-vector (*P. alienus*) was observed. The authors were able to pinpoint the molecular interaction between the virus and the host proteins involved in the accumulation of the virus in the sperm. Differences in the aminoacidic sequence of the virus target protein, an HSPG, may explain differences in the transmission efficiency of the virus in nature.

However, some aspects need further clarification:

Introduction

1. Vertical, transovarial, paternal/maternal transmission are used almost indistinguishable in the introduction and in some figures, I recommend the authors to clearly specify how these terms are used through the manuscript.
2. Line 47: Since an important number of arboviruses are not transmitted vertically among their main vectors, specific examples of the viruses vertically transmitted, specially plant viruses, is needed.

Results

3. Are uninfected females able to transmit the virus after an encounter with an infected male (venereal transmission)? What is the efficiency of such mode of transmission?
4. Are the offspring from maternal transmission equally able to reach adulthood and transmit the virus than the one from paternal transmission?

Methods

1. How insects (lines 257 – 259) were infected and how the infection was determined? This needs to be specified in the crossing experiment section.

Reviewer #3 (Remarks to the Author):

The work by Mao and co-workers demonstrates that Rice gall dwarf virus associates with the sperm of the green rice leafhopper vector to facilitate paternal transmission. The finding is novel, will be

interest to the arbovirus community and has important epidemiological implications. Importantly, the authors then go on to demonstrate the interacting parties in the sperm and the virus. They also find that this type of transmission occurs the minor vector, *Nephotettix cincticeps*. In general there is a large body of work here that is thorough, convincing and well executed. I have a few comments below that need to be addressed before publication is warranted.

Can figure 1d be explained further to show that “weeds” do not participate in transmission cycle, while rice does.

What is the relevance of the experiment depicted in figure 2C to the overall story of the paper? If relevant, can the significance of the work be either expanded upon

Lines 126 and 127 state that “viruliferous females died earlier”. I assume this is in comparison to non-viruliferous females. However, the data in 2b do not support this claim because no significance is noted between any data points in the graph. I see that the trend is there, but if the values are not significantly different then this should be stated in the text.

Also, is the use of letters after the means in the figures (2a, 2d; but really tables) conventional? Perhaps these can be altered slightly to make it clearer that these denote significance. At the moment they are quite inconspicuous.

Line 160-162: Can a positive control for detection of RGDV and P2 with antibodies toward P2 be provided in the supplement?

Lines 189-191: It seems obvious that virions infect offspring after the sperm fertilizes the oocyte, but no fertilized eggs are shown in the data. Have newly fertilized oocytes been looked at to provide evidence of this? These experiments would show irrefutable evidence of the mechanism for vertical transmission.

Lines 198-203: Please clarify the species of leafhopper being discussed in this sentence.

Line 517: “same column (a) or line (c)...” –should be line (d), not (c).

Fig 3: Should indicate what colours represent in the legend. Red, green, blue, yellow. Currently, the only explanation is given for red and blue with relation to panel c.

In the spirit of open access, I would encourage authors to make the data available using some type of repository, rather than upon request.

Does the abstract comply to the word limit for the journal?

Minor issues.

L46. "the Americas", not America.

L241. Even "though"

Reviewer #4 (Remarks to the Author):

In this paper the authors decipher the vertical transmission of a plant reovirus, the Rice gall dwarf virus or RGDV) transmitted by leafhoppers. The authors show that paternal transmission is more efficient than maternal transmission (which was already published before) and that it plays a major role for the virus maintenance on the winter host of the leafhopper. They give evidence that the paternal transmission has no detrimental effect on the eggs development and on the male fitness in contrast to the maternal transmission of the virus. They give a nice demonstration by immunofluorescence observations that the virus is transported from infected males into females with sperm being attached to sperm heads. They clearly demonstrate that the major capsid protein of RGDV (the P8 protein) is responsible for the binding to the sperm head. They perform a pull down assay with the P8 protein and identify the heparan sulfate proteoglycan (HSPG) as a potential partner of P8. The domain III of HSPG was shown to bind to P8 in yeast and in vitro. Expression of HSPG was overexpressed in the male reproductive system and expression was induced by virus infection. Knocking down the expression of HSPG encoding gene by microinjection of specific double-stranded RNA into the male vector reduces the vertical transmission of the virus. Using another leafhopper species that transmits less efficiently RGDV, the authors show that the paternal transmission of the virus is also more efficient than the maternal transmission. They perform virus

immunolabeling in a non-vector species and yeast two hybrid to evaluate binding ability of the minor capsid protein P8 with the HSPG of the vector and non-vector species.

Overall, this paper gather an enormous amount of experiments which probably also explains why all of them and not always precisely described. The results of the experiments give new data on the molecular mechanism that support paternal transmission of viruses and in this respect this paper is very informative for the community working on virus transmission by insects. I however have some concerns (major and minor) on the way some experiments were conducted or analyzed.

Please find below my comments:

Major concerns:

1. How the selection of the original parents in the different crosses shown in Figure 1 is done is not very clear. Are the crosses performed randomly without knowing whether the parents are viruliferous or not? It seems that the virus detection is performed after the crosses. Some information is present in the sup Mat and Met but never mentioned in the text. Figure 1 is probably the less clear figure in the manuscript. Authors mention "viruliferous rate or frequency" which I think is the same. What is the difference between 0, 1st, 2nd, (Figure 1E) and F1, F2, F3 (Figure 1G). In this Figure the authors measure the % of viruliferous insects (males and females) and the vertical transmission of the virus. Do they want to link the two set of data? They tend to show that paternal transmission is more efficient than maternal transmission (Figure 1B, C and G). However, paternal transmission means that the virus will be more efficiently transmitted from the infected males than the infected females but it does not mean that in the progeny more males will be infected. How do the authors explain why more males are infected (Figure 1F) and why the differences in the percentage of viruliferous insects between males and females tend to decrease during the crosses (compare 0 and 2nd) in Figure 1F?
2. I also have a major concern regarding the identification of the heparin sulfate proteoglycan protein (HSPG). The authors did not mention which band from the GST-pull down experiment (Figure S1) was extracted from the gel and sequenced. I think that an additional control using the viral P2 protein (not supposed to bind to HPSG) should be shown to show that the band that was extracted from the gel is specific to the P8 pull-down. Authors should also mention what is the difference between GST-P8-1 and -2?? A more precise description in the legend of Fig1S (B) should be given since FigS1B appears before Fig4E which describes the different domains of HSPG. The mass spectrometry technique used to identify the peptidoglycan is not very clear in the manuscript. L168: Authors mentioned that 10 different peptides were identified by mass spectrometry but in Fig1S we can see 12 peptides and only four different peptides. Clarify this point.
3. The interaction between P8 and the different domains of HSPG was addressed by yeast two hybrid but again some important controls are missing. Do authors have evidence that the difference domains of HSPG are expressed in yeast? In Figure 1F, authors should show whether the domain III of HSPG alone can self-activate the transcription of the reporter genes.

4. Information on the HSPG gene is missing. Is it a unique gene or does it belong to a multigenic family? This information is very important to analyze the RNA interference experiment. The size and sequence of the dsRNA used to knock down the gene should be mentioned.
5. Detection of HSPG in females should be mentioned in the manuscript.
6. Figure 5 is not well annotated and does not correspond to the text in the manuscript nor in the Mat and Met section. Therefore, it was difficult to evaluate this part of the work.

L206: the non-vector wheat leahopper *Psammotettix alienus* is mentioned in the text whereas in the figure data from *P. striatus* is shown. The legend is also confusing since the three species are mentioned.

L208: *P. alienus* is mentioned in the text whereas in the figure it is *P. striatus*. *P. striatus* is not mentioned before and we do not know whether it is a vector or a non-vector species. I guess from the results that it is a non-vector species. The number in the figure are also not well assigned. Figure 5D is not the amino acid sequence and the yeast two hybrid is 5F and not 5E. The authors mentioned 10 different amino acids between vector and non-vector species (as far as I understand with the problem of mixing names) but in the figure we see much more than 10 amino acids. In Figure 5D: the authors should compare similar mode of virus inoculation: microinjection of the virus in both case. They are comparing virus detection in viruliferous aphids for *N. cincticeps* with a virus microinjection in *P. striatus*.

Minor concerns

1. Authors did not refer to sup data materiel and methods. It is why it looks like some technical information is missing which can be found thereafter. RT-quantitative PCR lacks technical details (reference genes), number of individuals assayed, number of repeats.
2. Authors mentioned two antibodies anti DoIII of HSPG or anti HSPG. Are they the same or different antibodies. It is confusing.
3. Regarding the title, I am wondering whether “with” is the appropriate preposition after hitchhike. This structure is used in other locations in the manuscript.
4. Figure 1: as mentioned in the legend, each histogram represents the mean of 3 samples (N=3) which should represent 3 independent crosses of each combination. In the mat and methods section however it is mentioned 6 or 9 replicates and 3 repeats. What has really be done in each experiment? It should be clarified. I think that in the legend of the figure it should be mentioned that the eggs and not the larvae were analyzed (not obvious in the figure 1A).
5. It seems that after the crosses, the viruliferous status of the insects from the laboratory or caught in the field were assessed 5 days for males or 10 days for females after the crosses. Is there any possibility that virus contamination occurred after the mating and that one of the parents was originally not infected at the time of crossing? The procedure to select the parents for the crosses is

not clear and should be clarified. Concerning the procedure for the crosses (L264 and L267), can the authors mention why they replace daily the seedlings? Is it to avoid virus acquisition through the plant? Need to be clarified.

6. When authors analyzed the epidemiological significance of paternal transmission, I think that an important information is whether the weed *Alopecurus aequalis* is really a host for the virus or not. The sentence L108: "Generally, RGDV infection of the weed is not observed during winter" is vague and should be better clarified. The figure explaining the experimental set-up is also not very clear (Figure 1d): why is there two arrows starting from rice and going to the weeds? This inset should be better explained.

Other minor comments

L22: Infection of ovaries. This term "infection to" instead of "infection of" is used several times in the document

L25: Rice gall dwarf virus in italics with capital letter for Rice

L27: revealed that

L48: The authors used "cross-species transmission" for "horizontal transmission" which is more defined as a transmission within a species but between individual through the plant. I do not think therefore that "cross-species" is an appropriate term.

L51: this sentence is awkward

L53-54: the authors suggest that the sperm-mediated transmission only occurred (or has been observed) in controlled conditions? Clarify this point.

L71: there are other references that should be cited since there are other cases of plant virus transovarial transmission (some other reoviruses, tospoviruses, rhabdoviruses, tenuiviruses and orthobunyaviruses for example)

L79: has long been thought to be transmitted by a

L122: at least is a wrong term. Four is the maximum when considering Figure 2a.

Figure 2b and 2c: Need to mention the results of the statistical analysis in the figure.

L123: non-infected rather than un-infected

Figure 2c: transmission experiment not described in the manuscript. How many plants inoculated?

Legend of Figure 2: error: same column (a) or line (d) and not (c).

Figure 2d: the parental transmission had no effect on the production, development and hatching. It depends on which combination you are focusing on because if you compare the crossing between

viruliferous female and nonviruliferous male with that of nonviruliferous female and viruliferous male there are statistical differences.

Figure 3a-c: a negative control showing the immunofluorescence of a nonviruliferous male reproductive system would have been informative.

Figure 3g: if a labelling was supposed to be seen, if it is the case, it is not visible.

L140: what is “stimulation”? It is not described in the manuscript.

As shown in Table S2, authors never detected the virus in the oocytes of female after mating. Did they observe the virus at a later time? It should be interesting to mention it.

Figure 4A: mention the time of incubation below each inset.

Figure 4G: a pull down assay with other controls like the His-P2 or with the other domains of HSPG and the His-P8 would have been very informative.

Figure S2A: why the same band appears on the western blot considering that in lane 1 it is the domain III of HSPG expressed in *E. coli* and in lane 2 the whole HSPG protein from the insect?

Figure S4A, B: authors need to tell what is the antiserum against HSPG? Again is there one or two antisera against HSPG?

Figure 4J: mention in the legend that co-localization should be labeled in white.

Figure 4K needs additional control: incubation with the pre-immune serum or an unrelated antiserum

Table S3: mention that this table corresponds to the same results as the one presented in Figure 4N.

Figure 4M: why authors prefers to show a western blot to demonstrate the efficiency of the dsRNA to reduce expression of the HSPG protein and not an immunofluorescence assay as shown before in 4A, C, D, J and K?

Figure 4N: authors mentioned in the mat and methods to refer to a paper Chen et al (2013) for the design of dsRNA. In this paper the synthesis of the dsRNA targeting the HSPG is not described. What is the specificity of the dsRNA? Is the genome of *R. dorsalis* known? What is the stability of the reference genes used in RT-qPCR (ref genes are not mentioned) after microinjection of the dsRNA?

L194: minor vector of RGSV

Figure 5A: the viruliferous rate has been analyzed on both males and females? Specify

L203-206: rephrase

In the mat and methods section: this section should be carefully checked again and some missing technical information should be added like the quantity of virus, proteins or antibodies used in the *in vitro* binding assay.

L322: why “comprehensive” GST pull-down assay?

L323: perm lysates were prepared, but how? What is the composition of the buffer used to extract the proteins? Is it compatible with the extraction and solubilization of membrane proteins?

L650-651: How can a virus detection be done on males and females after mating and then the females left for oviposition? The virus detection method is destructive.

L688: authors mentioned two references for the preparation of HSPG polyclonal antibodies but no information is given in the two references. Maybe just delete the references.

This ref reporting a paternal transmission of a plant virus has not been mentioned in the manuscript: Zeigler RS, Morales FJ. 1990. Genetic determination of replication of rice hoja blanca virus within its planthopper vector, *Sogatodes oryzae*. *Phytopathology* 80:559–66

Annotation of figures is not homogeneous (see Y axis title in Figure 1b and c)

What is the proportion of males and females hatching from the eggs? This should be explained in the text since it can influence the viruliferous rate.

For all figures be homogeneous for capital or lower case letters

Name of insects in italics

Point-by-Point Responses to Reviewers' Comments

Reviewer #1:

This manuscript by Mao et al provided a definitive and mechanistic insight into one of the most perplexing and puzzling mechanisms of arbovirus transmission. While the subject has been known for some time now, and several proposed methods have been put forward over the years (most revealing experiments done with transovarial transmission of LaCrosse virus), this is the first manuscript that presents in an insightful and elegant way the mechanism by which paternal transmission occurs. Both methods of transmission (venereal as well as transovarial) have far reaching consequences since they have been proposed for many years as the most plausible mechanisms to explain the maintenance of arboviruses in the interepidemic years of transmission. This certainly may be a potential and highly effective method (paternal) of transmission that could also explain the transmission of insect specific arboviruses, a diverse group of viruses that have been discovered and described in some detail over the past 6-7 years.

It is quite remarkable that the model plant viruses used herein utilize heparin sulfate moieties to attach on the outer sperm surface, suggesting a conserved mechanism as HS has been identified as a receptor for rapid penetration of viruses, and bacteria on their target cells.

Overall this is a seminal and groundbreaking study that will impact the field of arbovirology for some years to come. One remaining aspect that this work barely touched is how frequent this mode of transmission is in nature which is urgently needed to inform dynamic models of transmission.

The manuscript is well written and can only require a handful of cosmetic corrections;

Response: We appreciate your positive evaluation of our study and your suggestion for us to improve the manuscript.

Overall this is a seminal and groundbreaking study that will impact the field of arbovirology for some years to come. One remaining aspect that this work barely touched is how frequent this mode of transmission is in nature which is urgently needed to inform dynamic models of transmission.

Response: We thank the reviewer for raising this issue. In Figure 1c, we measured the paternal and maternal transmission rates of RGDV in field-collected *Recilia dorsalis* populations. The paternal transmission rate (~60%) was evidently higher than the maternal rate (~20%), and in Figure 1f, more male insects were infected by RGDV than female insects, thus, we believe that paternal virus transmission occurs pretty frequently in nature. We also discuss this lines 239-244.

Line 215-216. Please cite the studies describing mechanisms of arbovirus maternal transmission

Response: We thank the reviewer for pointing out this problem. We added the references describing mechanisms of arbovirus maternal transmission in line 229 in revised version.

Reviewer #2:

Mao et al. introduce a novel concept and evidence for the paternal transmission of the RGDV and their impact on the virus and host survival as well as the epidemiological relevance of such mode of transmission. The authors show compelling evidence of vector-host interactions resulting in the paternal transmission of the RGDV as the most efficient method for its maintenance in nature. This transmission mechanism was also found in the minor RGDV vector *N. cincticeps* while no attachment of the virus to the sperm of a non-vector (*P. alienus*) was observed. The authors were able to pinpoint the molecular interaction between the virus and the host proteins involved in the accumulation of the virus in the sperm. Differences in the aminoacidic sequence of the virus target protein, an HSPG, may explain differences in the transmission efficiency of the virus in nature. However, some aspects need further clarification:

Response: Thanks for the positive and constructive comments from this reviewer.

Vertical, transovarial, paternal/maternal transmission are used almost indistinguishable in the introduction and in some figures, I recommend the authors to clearly specify how these terms are used through the manuscript.

Response: Generally, vertical virus transmission includes paternal and maternal transmission. Maternal transmission is generally mediated by transovarial transmission by female insects, however, whether male sperm-mediated paternal transmission occurs remain undetermined. We have revised to make it clear in lines 51-55 and also through the manuscript.

Line 47: Since an important number of arboviruses are not transmitted vertically among their main vectors, specific examples of the viruses vertically transmitted, specially plant viruses, is needed.

Response: Thank you for the advice. We added the example of rice stripe virus in lines 46-47.

Are uninfected females able to transmit the virus after an encounter with an infected male (venereal transmission)? What is the efficiency of such mode of transmission?

Response: In the revised manuscript, we tested whether uninfected females were able to transmit the virus after an encounter with an infected male (venereal transmission). As shown in Figure S1b, V⁺ females which acquired virus through mating with V⁺ male were able to transmit RGDV to rice plants, and the transmission efficiency varied from 0 to 27% depending on the days post mating. Related contents appeared in lines 157-159.

Are the offspring from maternal transmission equally able to reach adulthood and transmit the virus than the one from paternal transmission?

Response: Thank you for raising this issue. In the revised manuscript, we measured the development of the offspring from maternal and paternal transmission and their ability to transmit virus. As shown in Figure S1a and lines 132-133, the surviving offspring from maternal transmission were able to reach adulthood and transmit the virus as efficient as the ones from paternal transmission.

How insects (lines 257 – 259) were infected and how the infection was determined? This needs to be specified in the crossing experiment section.

Response: Thank you for this suggestion. We added more details to address how insects were infected and how the infection was determined in lines 280-296.

Reviewer #3:

The work by Mao and co-workers demonstrates that Rice gall dwarf virus associates with the sperm of the green rice leafhopper vector to facilitate paternal transmission. The finding is novel, will be interest to the arbovirus community and has important epidemiological implications. Importantly, the authors then go on to demonstrate the interacting parties in the sperm and the virus. They also find that this type of transmission occurs the minor vector, *Nephotettix cincticeps*. In general there is a large body of work here that is thorough, convincing and well executed. I have a few comments below that need to be addressed before publication is warranted.

Can figure 1d be explained further to show that “weeds” do not participate in transmission cycle, while rice does.

Response: In the revised manuscript, we have explained further the role of the weed in virus transmission cycle in lines 105-113 and Table S1.

What is the relevance of the experiment depicted in figure 2C to the overall story of the paper? If relevant, can the significance of the work be either expanded upon. Lines 126 and 127 state that “viruliferous females died earlier”. I assume this is in comparison to non-viruliferous females. However, the data in 2b do not support this claim because no significance is noted between any data points in the graph. I see that the trend is there, but if the values are not significantly different then this should be stated in the text.

Response: We appreciate this comment. Figure 2c depicts the effects of paternal virus transmission on the fitness of adult males or their offspring, thus, Figure 2c is not relevant with the topic. We have deleted this figure in the revised version. We added the statistical analysis for Figure 2b, and the significance was noted between the nonviruliferous and viruliferous females. We also added “compared with V⁻ females or V⁺ males” in lines 130-132.

Also, is the use of letters after the means in the figures (2a, 2d; but really tables) conventional? Perhaps these can be altered slightly to make it clearer that these denote significance. At the moment they are quite inconspicuous

Response: Yes, the use of letters after the means is conventional. Statistical significance was evaluated using one-way ANOVA at a 0.05 level followed by Tukey’s honestly significant difference (HSD) test for leafhopper performance. The same letter indicated no significance. In the figure legend, we also noted the different letters after means in the same column (a) or line (d) indicated a significant difference at $P = 0.05$.

Line 160-162: Can a positive control for detection of RGDV and P2 with antibodies toward P2 be provided in the supplement?

Response: Thank you for the advice. In the revised manuscript, we added specificity detection of antibodies against P2 and P8 in Figure S2.

Lines 189-191: It seems obvious that virions infect offspring after the sperm fertilizes the oocyte, but no fertilized eggs are shown in the data. Have newly fertilized oocytes been looked at to provide evidence of this? These experiments would show irrefutable evidence of the mechanism for vertical transmission.

Response: Thank you for pointing out this, we have tried to trace virus in the newly fertilized oocytes. However, once fertilized, the eggs were oviposited very soon, and we were unable to collect oocytes at this stage for virus detection. In terms of the short period interval between fertilization and oviposition, the embryo started to develop after fertilization. Thus, it's hard to trace virus in the newly fertilized oocytes. Usually, we directly picked up eggs from rice plants after oviposition for viral detection

Lines 198-203: Please clarify the species of leafhopper being discussed in this sentence.

Response: Thank you for raising this issue. In the revised version, we added the species of leafhoppers in lines 207-224.

Line 517: “same column (a) or line (c)...” –should be line (d), not (c).

Response: We have corrected.

Fig 3: Should indicate what colours represent in the legend. Red, green, blue, yellow. Currently, the only explanation is given for red and blue with relation to panel c.

Response: Thank you for raising this issue. The figure legends in the full text have been revised.

In the spirit of open access, I would encourage authors to make the data available using some type of repository, rather than upon request.

Response: Thank you for the advice. We have provided more data in Supplementary information in the revised version for assessing the manuscript.

Does the abstract comply to the word limit for the journal?

Response: We have revised the abstract to comply the word limit- **approximately** 150 words.

Minor issues.

L46. “the Americas”, not America.

L241. Even “though”

Response: We have corrected them in the revised manuscript.

Reviewer #4:

In this paper the authors decipher the vertical transmission of a plant reovirus, the Rice gall dwarf virus or RGDV) transmitted by leafhoppers. The authors show that paternal transmission is more efficient than maternal transmission (which was already published before) and that it plays a major role for the virus maintenance on the winter host of the leafhopper. They give evidence that the paternal transmission has no detrimental effect on the eggs development and on the male fitness in contrast to the maternal transmission of the virus. They give a nice demonstration by immunofluorescence observations that the virus is transported from infected males into females with sperm being attached to sperm heads. They clearly demonstrate that the major capsid protein of RGDV (the P8 protein) is responsible for the binding to the sperm head. They perform a pull down assay with the P8 protein and identify the heparan sulfate proteoglycan (HSPG) as a potential partner of P8. The domain III of HSPG was shown to bind to P8 in yeast and in vitro. Expression of HSPG was overexpress in the male reproductive system and expression was induced by virus infection. Knocking down the expression of HSPG encoding gene by microinjection of specific double-stranded RNA into the male vector reduces the vertical transmission of the virus. Using another leafhopper species that transmits less efficiently RGDV, the authors show that the paternal transmission of the virus is also more efficient than the maternal transmission. They perform virus immunolabeling in a non-vector species and yeast two hybrid to evaluate binding ability of the minor capsid protein P8 with the HSPG of the vector and non-vector species.

Overall, this paper gather an enormous amount of experiments which probably also explains why all of them and not always precisely described. The results of the experiments give new data on the molecular mechanism that support paternal transmission of viruses and in this respect this paper is very informative for the community working on virus transmission by insects. I however have some concerns (major and minor) on the way some experiments were conducted or analyzed. Please find below my comments:

Response: Thank you for the positive comments.

How the selection of the original parents in the different crosses shown in Figure 1 is done is not very clear. Are the crosses performed randomly without knowing whether the parents are viruliferous or not? It seems that the virus detection is performed after the crosses. Some information is present in the sup Mat and Met but never mentioned in the text. Figure 1 is probably the less clear figure in the manuscript. Authors mention “viruliferous rate or frequency” which I think is the same. What is the difference between 0, 1st, 2nd, (Figure 1E) and F1, F2, F3 (Figure 1G). In this Figure the authors measure the % of viruliferous insects (males and females) and the vertical transmission of the virus. Do they want to link the two set of data? They tend to show that paternal transmission is more efficient than maternal transmission (Figure 1B, C and G). However, paternal transmission means that the virus will be more efficiently transmitted from the infected males than the infected females but it does not mean that in the progeny more males will be infected. How do the authors explain why more males are infected (Figure 1F) and why the differences in the percentage of viruliferous insects

between males and females tend to decrease during the crosses (compare 0 and 2nd) in Figure 1F?

Response: Thank you for raising this point. We added more details in lines 280-296 to address how the selection of the original parents in the different crosses shown in Figure 1.

Viruliferous rate and frequency is the same, and we have replaced “frequency” with “rate”.

The “O”, “1st” and “2nd” in Figure 1e mean the original overwintering insects collected in November, the first generation of overwintering insects collected around January, and the second generation of overwintering insects collected in late February. The “F1, F2 and F3” in Figure 1g mean the 1st, 2nd and 3rd generation of offspring derived from paternal or maternal mating groups. Thus, these two figures are different. We have replaced F1, F2 and F3 with 1st, 2nd and 3rd in the figure and main text.

For the comment: “How do the authors explain why more males are infected (Figure 1F) and why the differences in the percentage of viruliferous insects between males and females tend to decrease during the crosses (compare 0 and 2nd) in Figure 1F?” In the Discussion, we added the sentences “Furthermore, viruliferous males survive much longer than viruliferous females, and finally, more males are infected by RGDV than females in the field. More importantly, the paternal transmission rate (~60%) is evidently higher than the maternal rate (~20%) in field-collected *R. dorsalis* populations. Thus, during the cold seasons unfavorable for virus-infected rice hosts in the field, the maintenance efficiency of RGDV through up to two generations tends to decrease” in lines 239-244 to explain it.

I also have a major concern regarding the identification of the heparin sulfate proteoglycan protein (HSPG). The authors did not mention which band from the GST-pull down experiment (Figure S1) was extracted from the gel and sequenced. I think that an additional control using the viral P2 protein (not supposed to bind to HSPG) should be shown to show that the band that was extracted from the gel is specific to the P8 pull-down. Authors should also mention what is the difference between GST-P8-1 and -2?? A more precise description in the legend of Fig1S (B) should be given since Fig1S1B appears before Fig4E which describes the different domains of HSPG. The mass spectrometry technique used to identify the peptidoglycan is not very clear in the manuscript. L168: Authors mentioned that 10 different peptides were identified by mass spectrometry but in Fig1S we can see 12 peptides and only four different peptides. Clarify this point.

Response: Thank you for pointing out these details. Figure S1a (Figure S3a in the revised manuscript) showed all the eluted proteins bound to GST or GST-P8, and we didn't extract specific protein band for MS analysis. All the proteins eluted from GST or GST-P8 conjugated Sepharose beads were sent for mass spectrometry analysis. GST-P8-1 and -2, as well as GST-1 and -2 are two replicates. We have clarified this in the Figure S3a and the legend. We also added more information of HSPG in the legend Figure S3b. Furthermore, in the Methods, we added more descriptions for mass spectrometry technique in lines 346-359.

As shown in Figure 4c and d, our incubation experiments indicated that it was P8 not P2 of RGDV that mediated the direct binding of RGDV virions to the sperm head, so we generated GST-fused P8 as a bait using GST pull-down and GST as a control. The P2 was not used in the GST-pull down experiment.

As shown in Supplementary Figure 3, we identified 12 peptides with four different types mapped to the domain III of HSPG. We modified the description in lines 178-179 in the revised manuscript.

The interaction between P8 and the different domains of HSPG was addressed by yeast two hybrid but again some important controls are missing. Do authors have evidence that the difference domains of HSPG are expressed in yeast? In Figure 4f, authors should show whether the domain III of HSPG alone can self-activate the transcription of the reporter genes.

Response: Thank you again for the advice. In the revised manuscript, we provided Western blot detection of the expression of HSPG domain I-V in yeast in Figure S4. The domain III of HSPG is not self-activating, and the evidence is shown in Figure 4f in the revised version.

Information on the HSPG gene is missing. Is it a unique gene or does it belong to a multigenic family? This information is very important to analyze the RNA interference experiment. The size and sequence of the dsRNA used to knock down the gene should be mentioned.

Response: Thank you for pointing out these problems. HSPG gene was a unique gene based on the transcriptome database of *R. dorsalis*, as shown in lines 360-363. We added the size of the dsRNA used to knock down the gene in lines 397-402 and the sequence has been uploaded in NCBI (GenBank accession number MH060173).

Detection of HSPG in females should be mentioned in the manuscript.

Response: Thank you for the suggestion. In the revised manuscript, as shown in Figure S5d and e, the expression level of HSPG in female reproductive system was detected using RT-qPCR and Western blot, and the results showed that HSPG expression level was higher in male reproductive system than in female reproductive system.

Figure 5 is not well annotated and does not correspond to the text in the manuscript nor in the Mat and Met section. Therefore, it was difficult to evaluate this part of the work.

L206: the non-vector wheat leahopper *Psammotettix alienus* is mentioned in the text whereas in the figure data from *P. striatus* is shown. The legend is also confusing since the three species are mentioned.

Response: Thank you for pointing out these problems. We re-annotated Figure 5 in lines 628-641 to make it clear and correspond to the text in the manuscript. The *P. striatus* is a type error, and we corrected it. We also rewrote the Methods about this part in Supplementary information in lines 131-149.

P. alienus is mentioned in the text whereas in the figure it is *P. striatus*. *P. striatus* is not mentioned before and we do not know whether it is a vector or a non-vector species. I guess from the results that it is a non-vector species. The number in the figure is also not well assigned. Figure 5D is not the amino acid sequence and the yeast two hybrid is 5F and not 5E. The authors mentioned 10 different amino acids between vector and non-vector species (as far as I understand with the problem of mixing names) but in the figure we see much more than 10 amino acids. In Figure 5D: the authors should compare similar mode of virus inoculation: microinjection of the virus in both case. They are comparing virus detection in viruliferous aphids for *N. cincticeps* with a virus microinjection in *P. striatus*.

Response: We appreciate the comments. The first comment is the same with the above last comment; we have replaced *P. striatus* with *P. alienus*. We have corrected all the figure numbers in the main text. Yes, more than 10 amino acids are different between the vector and the non-vector species; we have corrected it in the main text in line 221. We microinjected RGDV into both *N.cincticeps* and *P. alienus* as your suggested and the sperms were labeled with RGDV-rhodamine (red) and DAPI (blue), and the Figure 5d was replaced with immunofluorescence micrographs of sperms under microinjection treatment. The method was revised in Supplementary information in lines 131-149.

Authors did not refer to sup data materiel and methods. It is why it looks like some technical information is missing which can be found thereafter. RT-quantitative PCR lacks technical details (reference genes), number of individuals assayed, number of repeats.

Response: Thank you for raising the issues. In the revised manuscript, we added more details for RT-qPCR in main text in lines 418-421, and in Supplementary information in lines 123-125.

Authors mentioned two antibodies anti DoIII of HSPG or anti HSPG. Are they the same or different antibodies. It is confusing.

Response: They are the same antibody; we replaced anti DoIII of HSPG with anti HSPG in the revised manuscript.

Regarding the title, I am wondering whether “with” is the appropriate preposition after hitchhike. This structure is used in other locations in the manuscript.

Response: The manuscript has been proofread by a scientific editor; “with” is the appropriate preposition after hitchhike.

Figure 1: as mentioned in the legend, each histogram represents the mean of 3 samples (N=3) which should represent 3 independent crosses of each combination. In the mat and methods section however it is mentioned 6 or 9 replicates and 3 repeats. What has really be done in each experiment? It should be clarified. I think that in the legend of the figure it should be mentioned that the eggs and not the larvae were analyzed (not obvious in the figure 1A).

Response: We are sorry that our description was insufficiently clear. Each histogram represents the mean of 3 independent experiments, and for each independent experiment, about 6-9

independent crosses were conducted. We corrected it in the revised manuscript in lines 297-303. We deleted “N=3” in the figure legend, and the details that eggs were collected for RGDV detection were added in the figure legend.

It seems that after the crosses, the viruliferous status of the insects from the laboratory or caught in the field were assessed 5 days for males or 10 days for females after the crosses. Is there any possibility that virus contamination occurred after the mating and that one of the parents was originally not infected at the time of crossing? The procedure to select the parents for the crosses is not clear and should be clarified. Concerning the procedure for the crosses (L264 and L267), can the authors mention why they replace daily the seedlings? Is it to avoid virus acquisition through the plant? Need to be clarified.

Response: Thank you for raising the issues. Indeed, the males mated with females for 5 days, and the females were kept for another 5 days for laying eggs. The seedling was replaced daily with a new one to avoid virus acquisition through the plant. Furthermore, we have provided more information about the original parent insect selection. The clear descriptions of crossing experiment appeared in lines 280-296 in the revised version.

When authors analyzed the epidemiological significance of paternal transmission, I think that an important information is whether the weed *Alopecurus aequalis* is a really a host for the virus or not. The sentence L108: “Generally, RGDV infection of the weed is not observed during winter” is vague and should be better clarified. The figure explaining the experimental set-up is also not very clear (Figure 1d): why is there two arrows starting from rice and going to the weeds? This inset should be better explained.

Response: Thank you for raising issue. As revealed in Table S1, we have measured the infection rate of RGDV in the weed *Alopecurus aequalis* collected in the field, and none of weed plants were RGDV positive. However, under laboratory condition, very low rate of viral infection to the weed *Alopecurus aequalis* can be observed. Thus, we believe that the overwintering insects mainly inherit virus from their parents, not from the weeds. We have revised Figure 1d to make it clearer and explained in detail in lines 105-113.

Other minor comments

Response: Thank you for your kindness and patience to point out these details. The response followed by the comment.

L22: Infection of ovaries. This term “infection to” instead of “infection of” is used several times in the document

Response: We corrected it.

L25: Rice gall dwarf virus in italics with capital letter for Rice

Response: We corrected it.

L27: revealed that

Response: We corrected it.

L48: The authors used “cross-species transmission” for “horizontal transmission” which is more defined as a transmission within a species but between individual through the plant. I do not think therefore that “cross-species” is an appropriate term.

Response: We deleted cross-species transmission.

L51: this sentence is awkward

Response: We have revised and make the logic description.

L53-54: the authors suggest that the sperm-mediated transmission only occurred (or has been observed) in controlled conditions? Clarify this point.

Response: We didn't suggest that the sperm-mediated transmission only occurred (or has been observed) in controlled conditions, to make it clear, we deleted “in nature”.

L71: there are other references that should be cited since there are other cases of plant virus transovarial transmission (some other reoviruses, tospoviruses, rhabdoviruses, tenuiviruses and orthobunyaviruses for example)

Response: We added three more references.

L79: has long been thought to be transmitted by a

Response: We corrected it.

L122: at least is a wrong term. Four is the maximum when considering Figure 2a.

Response: We replaced “at least” with “around”.

Figure 2b and 2c: Need to mention the results of the statistical analysis in the figure.

Response: We added the results of statistical analysis in Figure 2b; there is no significance in Figure 2c.

L123: non-infected rather than un-infected

Response: We replaced un-infected with non-infected in the main text.

Figure 2c: transmission experiment not described in the manuscript. How many plants inoculated?

Response: Following the third reviewer's comment, as the Figure 2c is not relevant to the story in Figure 2, thus we deleted it.

Legend of Figure 2: error: same column (a) or line (d) and not (c).

Response: We corrected it.

Figure 2d: the parental transmission had no effect on the production, development and hatching. It depends on which combination you are focusing on because if you compare the crossing between viruliferous female and nonviruliferous male with that of nonviruliferous female and viruliferous male there are statistical differences.

Response: Thank you for pointing out this problem. We focused on the comparison between the crossing between nonviruliferous female and viruliferous male with that of nonviruliferous female and nonviruliferous male, and there are no statistical differences. We edited it in the main text in lines 127-130 to make it precise.

Figure 3a-c: a negative control showing the immunofluorescence of a nonviruliferous male reproductive system would have been informative.

Response: We added it in Figure 3a.

Figure 3g: if a labelling was supposed to be seen, if it is the case, it is not visible.

Response: The Figure 3g showed that RGDV did not infect the ovary at an early stage after mating with viruliferous males.

L140: what is "stimulation"? It is not described in the manuscript.

Response: We deleted this word.

As shown in Table S2, authors never detected the virus in the oocytes of female after mating. Did they observe the virus at a later time? It should be interesting to mention it.

Response: Thank you for the suggestion. We have detected virus in the oocytes after mating at 12 days as revealed in Supplementary Table 3, and we observed virus signal in one tested unfertilized oocyte by immunofluorescence. The female adults usually died after mating at 12 days, and we were unable to observe at days after 12 days. For the fertilized oocytes, once fertilized, the eggs were oviposited very soon, thus we were unable to collect oocytes at this stage for virus detection. Usually, we directly collected eggs from rice plants after oviposition for viral detection.

Figure 4A: mention the time of incubation below each inset.

Response: We added incubation time below each inset.

Figure 4G: a pull down assay with other controls like the His-P2 or with the other domains of HSPG and the His-P8 would have been very informative.

Response: Thank you for raising this point. We added the His-P2 as control and replaced Figure 4G.

Figure S2A: why the same band appears on the western blot considering that in lane 1 it is the domain III of HSPG expressed in *E. coli* and in lane 2 the whole HSPG protein from the insect?

Response: Sorry that we provided the incorrect figure; the two same bands are two replicates of domain III of HSPG expressed in male *R. dorsalis* adults. We corrected it and added the detection of domain III of HSPG expressed in *E. coli* in Figure S5a.

Figure S4A, B: authors need to tell what is the antiserum against HSPG? Again is there one or two antisera against HSPG?

Response: There is only one antiserum against HSPG.

Figure 4J: mention in the legend that co-localization should be labeled in white.

Response: We added white triangles to label the co-localization.

Figure 4K needs additional control: incubation with the pre-immune serum or an unrelated antiserum.

Response: We used RGDV+3% BSA as control, and we added the pre-immune serum treatment in revised manuscript in Figure 4k.

Table S3: mention that this table corresponds to the same results as the one presented in Figure 4N.

Response: We deleted this table in the revised manuscript.

Figure 4M: why authors prefers to show a western blot to demonstrate the efficiency of the dsRNA to reduce expression of the HSPG protein and not an immunofluorescence assay as shown before in 4A, C, D, J and K?

Response: Thank you for your advice. We added immunofluorescence assay about the expression of the HSPG protein in sperms after dsRNA treatment in Figure 4l.

Figure 4N: authors mentioned in the mat and methods to refer to a paper Chen et al (2013) for the design of dsRNA. In this paper the synthesis of the dsRNA targeting the HSPG is not described. What is the specificity of the dsRNA? Is the genome of *R. dorsalis* known? What is the stability of the reference genes used in RT-qPCR (ref genes are not mentioned) after microinjection of the dsRNA?

Response: We deleted this reference in the revised manuscript. HSPG gene was obtained depending on the leafhopper full-length transcriptome. We have described the synthesis of dsRNA targeting the domain III sequence of the HSPG gene. The specificity of the dsRNA has been tested by RT-qPCR and Western blot. The internal control gene *elongation factor 1alpha* gene (EF1, accession number AB836665) was used as ref gene and the information was added in lines 418-419 in main text and lines 123-124 in Supplementary information.

L194: minor vector of RGSV

Response: We have corrected this.

Figure 5A: the viruliferous rate has been analyzed on both males and females? Specify

Response: Yes, we analyzed the same number of males and females, as described in Supplementary information in line 134 and the figure legend.

L203-206: rephrase

Response: We have rephrased this sentence.

In the mat and methods section: this section should be carefully checked again and some missing technical information should be added like the quantity of virus, proteins or antibodies used in the *in vitro* binding assay.

Response: We have added the information for quantity of virus, proteins or antibodies used in the mat and methods section including the *in vitro* binding assay.

L322: why “comprehensive” GST pull-down assay?

Response: We deleted this word in the revised manuscript.

L323: sperm lysates were prepared, but how? What is the composition of the buffer used to extract the proteins? Is it compatible with the extraction and solubilization of membrane proteins?

Response: We added the description for sperm lysates preparation in lines 343-346. In the experiment, the sperm was suspended in PBS and lysed by supersonic wave, and then all the supernatants after centrifugation were used for GST pull-down. We didn't use a specific buffer to extract membrane protein, but it seems the membrane protein HSPG is soluble in PBS in this experiment.

L650-651: How can a virus detection be done on males and females after mating and then the females left for oviposition? The virus detection method is destructive.

Response: Thank you for raising this issue. We provided more details for virus detection in this experiment in lines 280-296 in Supplementary information. Briefly, the original viruliferous males or females were selected from the viruliferous leafhopper population kept in laboratory, then the males were collected after mating and detected using RT-PCR. The females were left to lay eggs for 8 days and then collected for RGDV detection using RT-PCR.

L688: authors mentioned two references for the preparation of HSPG polyclonal antibodies but no information is given in the two references. Maybe just delete the references.

Response: We have deleted the references.

This ref reporting a paternal transmission of a plant virus has not been mentioned in the manuscript: Zeigler RS, Morales FJ. 1990. Genetic determination of replication of rice hoja blanca virus within its planthopper vector, *Sogatodes oryzoicola*. *Phytopathology* 80:559–66

Response: We have carefully read this paper; it described the ability of insect progenies, derived from cross combinations: nonvector × nonvector, nonvector × vector, vector × vector, to transmit virus to plants. They didn't mention paternal or maternal virus transmission. Thus, it's not related to our study, and we didn't cite it.

Annotation of figures is not homogeneous (see Y axis title in Figure 1b and c)

Response: Thank you for pointing out these problems. We corrected it in revised manuscript.

What is the proportion of males and females hatching from the eggs? This should be explained in the text since it can influence the viruliferous rate.

Response: Thank you for raising this point. The sex ratio of offspring was not affected by RGDV vertical transmission pathway. In the revised manuscript, we added the information about the sex ratio of offspring in line 134-135 and Figure 2c.

For all figures be homogeneous for capital or lower case letters

Response: Thank you for pointing out these problems. We corrected them in revised manuscript.

Name of insects in italics

Response: We corrected them in revised manuscript.

REVIEWERS' COMMENTS:

Reviewer #1 (Remarks to the Author):

The authors have successfully addressed my concerns.

Reviewer #2 (Remarks to the Author):

The authors have responded to the comments satisfactorily. I recommend the acceptance of the manuscript. This is a highly relevant study addressing a gap in the knowledge of arthropod-pathogen interactions and transmission strategies.

Reviewer #3 (Remarks to the Author):

The authors have responded appropriately to the reviews. I have no further comments.

Reviewer #4 (Remarks to the Author):

The authors addressed all of my comments and performed the requested modifications. They not only gave answers to the questions but also performed additional experiments and included the results in figures. I just observed minor errors when reading the modifications introduced in the manuscript.

Line 475 : was replicated three times

Line 475: bound proteins

Line 69: Rice stripe virus: first letter uppercase and in italics

Line 179: as efficiently as

Line 203: indicating that strong

Line 206: increased with??

Line 368: mating one on one: is it the correct term?

Line 458: supersonic wave?? Isn't it more "ultrasounds"?

Point-by-Point Responses to Reviewers' Comments

Reviewer #4:

The authors addressed all of my comments and performed the requested modifications. They not only gave answers to the questions but also performed additional experiments and included the results in figures. I just observed minor errors when reading the modifications introduced in the manuscript.

Response: We appreciate the reviewer's patience to help improve the manuscript.

Line 475: was replicated three times

Response: We corrected it.

Line 475: bound proteins

Response: We corrected it.

Line 69: Rice stripe virus: first letter uppercase and in italics

Response: We have corrected it and revised the same problem throughout the ms.

Line 179: as efficiently as

Response: Thank you for the advice. We have replaced "as efficient as" with "as efficiently as".

Line 203: indicating that strong

Response: We have added "that" after "indicating".

Line 206: increased with??

Response: We have replaced "increased by" with "increased with".

Line 368: mating one on one: is it the correct term?

Response: We have replaced "mating one on one" with "mating one to one" in the revised version.

Line 458: supersonic wave?? Isn't it more "ultrasounds"?

Response: We have replaced "supersonic wave" with "ultrasounds" in the revised version.